# Online Multi-Class Selection with Group Fairness Guarantee

**Faraz Zargari**
University of Alberta
fzargari@ualberta.ca

**Hossein Nekouyan**
University of Alberta
nekouyan@ualberta.ca

**Lyndon Hallett**
University of Alberta
lhallett@ualberta.ca

**Bo Sun**
University of Ottawa, Vector Institute
bo.sun@uottawa.ca

**Xiaoqi Tan**
University of Alberta
xiaoqi.tan@ualberta.ca

## Abstract

We study the online multi-class selection problem with group fairness guarantees, where limited resources must be allocated to sequentially arriving agents. Our work addresses two key limitations in the existing literature. First, we introduce a novel lossless rounding scheme that ensures the integral algorithm achieves the same expected performance as any fractional solution. Second, we explicitly address the challenges introduced by agents who belong to multiple classes. To this end, we develop a randomized algorithm based on a relax-and-round framework. The algorithm first computes a fractional solution using a resource reservation approach—referred to as the *set-aside* mechanism—to enforce fairness across classes. The subsequent rounding step preserves these fairness guarantees without degrading performance. Additionally, we propose a learning-augmented variant that incorporates untrusted machine-learned predictions to better balance fairness and efficiency in practical settings.

## 1 Introduction

Online fair allocation has attracted increasing attention in recent years, as addressing algorithmic bias in decision-making has become a major concern in artificial intelligence [BMM24, MX24, CCDNF21a, SJB+23, HJS+24, GNPS24, HHIS23, BGH+23, HLSW23, BGGJ22]. Most existing work considers settings where agents are offline and resources arrive sequentially, requiring allocation strategies that maintain fairness across agents.

In contrast, this paper focuses on algorithmic fairness in the *online selection* problem, where the roles are reversed: agents arrive sequentially, and the decision-maker must allocate limited, offline resources by either accepting or rejecting each request immediately and irrevocably. This setting introduces unique algorithmic challenges for fairness, as the decision-maker must strategically reserve resources despite uncertainty about future arrivals. Despite its practical relevance, fairness in online selection with sequential agent arrivals has received comparatively little attention in the literature.

Recognizing the algorithmic challenges of ensuring fairness among online agents, a recent study [ZJST25] introduces a weaker fairness notion known as *group fairness*. In this model, each agent belongs to a single group (or class), and the algorithm aims to ensure fairness across groups rather than individual agents, thereby expanding the space of feasible fairness guarantees. However, the model in [ZJST25] has two key limitations. First, it focuses on *fractional allocation* (i.e., resources are divisible), whereas many real-world online allocation problems are inherently integral (e.g., allocation of public houses in social housing programs). Second, it assumes that each agent belongs to exactly one group, while in practice agents are often *multi-labeled*—i.e., simultaneously associated

39th Conference on Neural Information Processing Systems (NeurIPS 2025).

with multiple groups (e.g., by region, gender, etc.). Ensuring fairness across such overlapping groups introduces additional complexity and interdependence.

Motivated by these limitations, this paper introduces and studies the Online Multi-class Selection (OMcS) problem with group fairness guarantees. In OMcS, online agents may belong to one or more of a fixed number of classes (or groups), and the goal is to maintain certain fairness notions across these groups without prior knowledge of the number of agents in each group. Many real-world applications fall under this setting and raise fairness concerns. For example, in cloud job scheduling [ZLW17, ZHW$^+$15], the system must allocate limited CPU (or GPU) resources fairly across jobs (i.e., agents), which may be multi-labeled based on user type, geographic region, and other attributes. Ensuring fairness across such overlapping groups is critical for mitigating unequal access to public computing power. In this work, we aim to design algorithms that achieve optimal group fairness guarantees in OMcS. This objective is particularly challenging in the multi-labeled setting, where accepting a single agent can simultaneously enhance the utility of multiple classes, thereby complicating the task of maintaining the desired fairness guarantees.

## 1.1 Our Contributions

We examine OMcS under two fairness criteria: Group Fairness by Quantity (GFQ) and $\beta$-Proportional Fairness ($\beta$-PF). Under GFQ, we require that each group receive a fixed quota of resources. For any GFQ specification, we present two algorithms: an optimal deterministic algorithm (Theorem 3.1), and a randomized algorithm that applies a lossless rounding scheme to any optimal solution in the fractional setting (Theorem 3.2). Our randomized approach operates within a relax-and-round framework and introduces a novel lossless online rounding scheme. For the $\beta$-PF objective, we first develop an optimal fractional solution specifically designed for the multi-labeled setting (Theorem 4.1), and then convert it into an integral allocation via a lossless rounding procedure that preserves the fractional performance guarantee exactly (Theorem 4.3).

Although our proposed algorithm achieves the optimal fairness guarantee, the design based on worst-case analysis is often too pessimistic for practical applications. To mitigate this, we leverage the learning-augmented algorithm framework by incorporating black-box machine-learned advice from a fair online allocation. This algorithm can significantly improve fairness when the advice is fair (i.e., consistency) while still ensuring a worst-case fairness guarantee even if the advice is entirely unfair (Theorems 5.1). Technically, our main contribution is a *lossless online rounding scheme* inspired by the lossless online correlated $k$-rental scheme introduced in [NSBT25], which converts any fractional solution into an integral one without any expected performance loss. Beyond OMcS, this scheme is of independent interest and can be applied broadly to online selection and revenue-management problems, closing the integrality gap and yielding tight bounds under fairness constraints.

## 1.2 Related Work

Online selection problem has been extensively studied under various assumptions about arrival sequences, including the random order model in the secretary problem [Gar70, AMW01, CCDNF21b], the IID arrivals in the prophet inequality [CFH$^+$19, SC84], and adversarial arrivals in online search problems [LPS09, JLTZ21]. In this paper, the OMcS framework builds upon the adversarial model. Below, we briefly review the most relevant works to our study.

**Online selection.** Adversarial online selection problem assumes the valuations of online arrivals are bounded within a finite support. Under this assumption, the classic $k$-search problem introduced in [LPS09, EYFKT01] and the online knapsack problem in [ZCL08, CZL08] developed threshold-based algorithms that can achieve optimal worst-case performance under competitive analysis. [TYBLG25] applied similar approaches in the setting of online selection with convex costs, demonstrating the optimality of their approach for large-inventory scenarios and asymptotic optimality for small-inventory cases. Despite these advancements, most existing works heavily rely on the large inventory assumption in their analysis, and it is inherently challenging to design algorithms for online selection problems in small inventory settings with tight performance bounds.

**Online rounding.** The online rounding framework has recently attracted significant attention from both computer science [FHTZ22] and operations research [Ma24a]. These rounding schemes primarily rely on the relax-and-round approach. Specifically, in the first step, the problem is relaxed and formulated as a linear program. In the second step, the solution is rounded to construct a computationally efficient online decision-making policy [BN09, BBMN15, CPW19]. Huang et al. [HZZ20]

introduced the Online Correlated Selection (OCS) algorithm for the online matching problem. This algorithm belongs to the class of randomized rounding approaches and establishes negative correlations between sequential decisions. Subsequently, Fahrbach et al. [FHTZ22] introduced an OCS-based algorithm for edge-weighted online bipartite matching, and Huang et al. [HZZ24] adapted similar techniques for the Adwords problem. Furthermore, a recent work [NSBT25] develops a new online rounding scheme, called the online correlated $k$-rental scheme, for online selection problems with reusable resources. This scheme losslessly rounds fractional solutions into integral decisions using a single random seed sampled at the beginning of the algorithm.

**Online fair allocation.** Fairness in resource allocation has been a key area of research in computer science, operations research, and economics, resulting in a wide range of studies on the equitable distribution of divisible and indivisible resources (e.g., [Ste48, DS61]). [HHIS23] investigated class fairness in online bipartite matching using the concept of *envy-freeness up to one item*, and Banerjee et al. [BGH$^+$23] focused on proportional fairness in fractional online matching. Other works, such as [HLSW23, BGGJ22], explored the maximization of Nash social welfare in similar settings. However, these studies assume offline agents with online resource arrivals, neglecting scenarios with sequential agent arrivals. Our work addresses this gap by examining settings where resources are offline and agents arrive sequentially. This scenario poses unique challenges, as irrevocable allocations can disadvantage future agents given fixed resources. Closest to our setting, [JZST24] explores quantity-based fairness in fractional online allocation. Quantity-based fairness, relying on predefined criteria, complicates analyzing fairness-efficiency trade-offs. Recently, [ZJST25] extended this to utility-based fairness in the fractional setting; however, extending these fractional results to integral allocations with multi-labeled arrivals remains significantly challenging.

## 2 Problem Formulation and Preliminaries

In this section, we introduce and formulate the online multi-class selection problem, and formalize the notation of efficiency and group fairness in this paper.

### 2.1 OMcS: Problem Statement and Assumptions

We consider an online multi-class selection problem (OMcS) defined as follows: A seller has an initial inventory of $B$ units of indivisible resources to allocate to a sequence of agents arriving one at a time. Upon the arrival of agent $t \in [T]$, the agent submits a request for one unit of the resource along with a valuation $v_t$ (i.e., their willingness to pay). The seller must make an immediate and irrevocable binary decision $x_t \in \{0, 1\}$: setting $x_t = 1$ indicates accepting the offer and allocating one unit of the resource; $x_t = 0$ indicates rejection. In OMcS, each agent $t$ is associated with a label set $\mathcal{J}_t \subseteq [K]$, representing the classes to which the agent belongs. If $|\mathcal{J}_t| = 1$, we refer to agent $t$ as a ***single-labeled*** agent; if $|\mathcal{J}_t| > 1$, the agent is ***multi-labeled***.

In the OMcS problem, we assume that the valuations of agents in each class $j \in [K]$ are bounded within the interval $[1, \theta_j]$, where $\theta_j$ is referred to as the *fluctuation ratio* of class $j$. For agents belonging to multiple classes, their valuations are bounded by $[1, \min_{j \in \mathcal{J}_t} \theta_j]$. A larger $\theta_j$ indicates greater variability in valuations within the class, while a smaller $\theta_j$ implies more uniformity. Without loss of generality, we assume $\theta_1 \leq \theta_2 \leq \cdots \leq \theta_K$.

A commonly studied special case in the online selection literature [EYFKT01, LPS09, JLTZ21, SLH$^+$21, TYBLG25] assumes a universal fluctuation ratio, i.e., $\theta_j = \theta$ for all $j \in [K]$. Such interval bounds may be adopted as standard modeling assumptions or derived from trusted predictions [JLTZ21, HS25]. We assume that the initial inventory $B$, the number of classes $K$, and the fluctuation ratios $\{\theta_j\}_{j \in [K]}$ are known a priori, while all other information–including the valuations $\{v_t\}_{t \in [T]}$, the total number of arrivals $T$, and the label sets $\{\mathcal{J}_t\}_{t \in [T]}$–remains unknown.

### 2.2 Efficiency and Fairness Metrics

We consider the following performance metrics to evaluate the efficiency and fairness of online algorithms for OMcS.

**Efficiency metrics: competitive ratio in *utility* maximization.** A seller's primary goal is to maximize the total utility of all agents regardless of their groups, i.e., $\sum_t v_t x_t$, subject to the resource constraint $\sum_t x_t \leq B$. For a given arrival instance $I = \{(v_1, \mathcal{J}_1), (v_2, \mathcal{J}_2), \ldots, (v_T, \mathcal{J}_T)\}$, let $\mathsf{OPT}(I)$ represent the optimal achievable total utility in the offline setting, where the sequence $I$ is

known in advance. $\mathsf{OPT}(I)$ can be determined by solving the following integer program:

$$\mathsf{OPT}(I) = \max_{x_t \in \{0,1\}} \sum_t v_t x_t, \quad \text{s.t.} \sum_t x_t \leq B. \tag{1}$$

In the online setting, we use the *competitive ratio* as our efficiency metric. Let $\mathsf{ALG}(I)$ be the revenue of an online algorithm $\mathsf{ALG}$. The goal is to minimize the worst-case competitive ratio, defined as $\mathsf{CR}^* := \min_{\mathsf{ALG}} \max_{I \in \Omega} \frac{\mathsf{OPT}(I)}{\mathbb{E}[\mathsf{ALG}(I)]}$, where $\Omega$ is the set of all possible arrival sequences within the intervals characterized by $\{\theta_j\}_{j \in [K]}$, and $\mathsf{OPT}(I)$ is the offline optimal revenue.

**Fairness metrics: group proportionality by *quantity* and *utility*.** We focus on two fairness metrics. The first one is *quantity-based fairness*, which requires that a minimum amount of resources be allocated to agents from each class. This notion, referred to as *Group Fairness by Quantity*, is formally defined as follows:

**Definition 1** (Group Fairness by Quantity (GFQ)). An allocation $\mathbf{x} := [x_1, \ldots, x_T]$ satisfies group fairness by quantity if $\sum_{t \in [T]} x_t \cdot \mathbf{1}_{\{j \in \mathcal{J}_t\}} \geq m_j$ holds for all $j \in [K]$, where $\mathbf{m} := \{m_j\}_{j \in [K]}$ is a pre-determined fairness requirement.

Under the multi-label setting, an agent with multiple labels can simultaneously help satisfy the GFQ constraints for several groups. However, once these quantity-based constraints are fulfilled, the seller's decision depends solely on valuations, and labeling information is ignored. This approach overlooks fairness during the allocation process and may not suit real-world applications where fairness must be maintained throughout. To address this, we introduce a *utility-based fairness* metric that evaluates fairness based on agents' utilities. This encourages the seller to favor multi-labeled agents, enhancing both overall utility while promoting a more balanced allocation across groups.

**Definition 2** ($\beta$-Proportional Fairness ($\beta$-PF)). Let utility of class $j$ with allocation $\mathbf{x} := [x_1, \ldots, x_T]$ be $U_j(\mathbf{x}) = \sum_{t \in [T]} v_t \cdot x_t \cdot \mathbf{1}_{\{j \in \mathcal{J}_t\}}$, where $\mathbf{1}_{\{j \in \mathcal{J}_t\}}$ is an indicator function. For $\beta \geq 1$, an allocation $\mathbf{x}$ is $\beta$-proportionally fair if, for every other allocation $\mathbf{w}$, the following inequality holds: $\frac{1}{K} \sum_{j \in [K]} \frac{U_j(\mathbf{w})}{U_j(\mathbf{x})} \leq \beta$.[1]

An online algorithm is said to be $\beta$-proportionally fair if it consistently produces allocations that satisfy $\beta$-PF under all possible arrival instances. When $\beta = 1$, the allocation is referred to as *proportional fair* and has been widely used in network resource allocation (e.g., [CFLM19, KMT98, KDRU16, Kel97]) and fair clustering algorithms (e.g., [CMS24, LLS$^+$21, MS20, CFLM19]). However, in online settings, due to future uncertainties, exact 1-PF is generally unattainable. Thus, we focus on its $\beta$-approximation, called $\beta$-PF [BGH$^+$23, MS20]. This fairness notion is widely used in the literature and it is closely related to the Nash Social Welfare (NSW). Specifically, if an algorithm is $\beta$-PF, it always produces $\beta$-NSW.

## 3  OMcS **with Group Fairness by Quantity**

In this section, we investigate OMcS under the GFQ constraints. Based on Definition 1, a reserved allocation $\mathbf{m}$ is provided in advance. Therefore we can reformulate the problem in (1) by adding a new set of GFQ constraints as $\sum_{t \in [T]} x_t \cdot \mathbf{1}_{\{j_t = j\}} \geq m_j$ for all $j \in [K]$. We aim to design an algorithm to maximize the efficiency (i.e., minimizing the competitive ratio) for a given GFQ requirement $\mathbf{m}$.

### 3.1  Warm Up: An Optimal Deterministic Set-Aside Algorithm

We begin by presenting a simple deterministic algorithm, termed D-SETASIDE-GFQ, for OMcS under GFQ constraints and show that it is optimal among all deterministic algorithms. D-SETASIDE-GFQ is a threshold-based algorithm and works as follows: upon receiving the first $m_j$ agents from each class $j \in [K]$, D-SETASIDE-GFQ ensures the corresponding fairness guarantee for that class, by automatically accepting these agents regardless of their requested valuation, until the fairness guarantee for the class is met. As a result, $M = \sum_{j \in [K]} m_j$ units out of $B$ resource items are reserved, or ***set-aside***, to meet the fairness requirements (hence the term 'set-aside' in D-SETASIDE-GFQ). The remaining $B - M$ items are then allocated to the arriving agents based on a *threshold*, denoted by $\boldsymbol{\lambda} = \{\lambda_i\}_{i \in [B-M]}$, where $\lambda_i$ denotes the threshold when $i$ units have been allocated. More

---

[1]We assume the fraction $x/y$ for non-negative $x$ and $y$ is equal to 0 when $x = y = 0$, while $x/y = +\infty$ when $y = 0$ but $x > 0$.

specifically, upon the arrival of an agent at time $t$, the algorithm first verifies whether the GFQ constraints for all associated classes are satisfied. If any of these constraints remain unmet, the agent is accepted unconditionally, regardless of its valuation. Otherwise, the agent is accepted only if its value exceeds the threshold for allocation at time $t$; if not, the agent is rejected. Let $C_j = B - \max_{i \in [j-1]}\{m_i\}$ and $D_j = \sum_{i=1}^{j-1}[m_i - \max_{l \in [i-1]}\{m_l\}]^+ \cdot \theta_i$, where $[\cdot]^+ = \max\{\cdot, 0\}$. In the following theorem, we formally present our design of the optimal threshold $\boldsymbol{\lambda}^*$ and the competitive ratio associated with it.

**Theorem 3.1** (OMcS with GFQ: Optimal Deterministic Algorithm). D-SETASIDE-GFQ *achieves the optimal competitive ratio of among all deterministic algorithms, denoted by $\alpha^*$, if and only if the threshold $\boldsymbol{\lambda}^* = \{\lambda_0^*, \lambda_1^*, \ldots, \lambda_\tau^*, \ldots, \lambda_{B-M}^*\}$ is designed as follows:*

*(i) If $\frac{B}{\alpha^*} \geq M$: the thresholds are split into two parts*

- $\lambda_0^* = \lambda_1^* = \ldots = \lambda_\tau^* = 1$ *and* $\lambda_{B-M}^* = \theta_K$, *where $\tau$ is the minimum integer in $\{0, 1, \ldots, B - M - 1\}$ such that $\tau + 1 \geq \frac{B}{\alpha^*} - M$.*

- $\{\alpha^*, \lambda_{\tau+1}^*, \ldots, \lambda_{B-M-1}^*\}$ *is the unique set of $B - M - \tau + 1$ positive real numbers that satisfy the system of equations:*

$$\alpha^* = \frac{\Delta^{\tau+1}}{\tau+1} = \frac{\Delta^{i+1} - \Delta^i}{\lambda_i^*} \quad \forall i \in [\tau+1, B-M-1],$$

  *where for some $\lambda_i^* \in [\theta_{j-1}, \theta_j]$, $\Delta^i = C_j \cdot \lambda_i^* + D_j$.*

*(ii) If $\frac{B}{\alpha^*} < M$: In this case, $\lambda_{B-M}^* = \theta_K$ and $\{\alpha^*, \lambda_0^*, \ldots, \lambda_{B-M-1}^*\}$ is the unique set of $B - M + 1$ positive real numbers that satisfy the system of equations:*

$$\alpha^* = \frac{\Delta^0}{M} = \frac{\Delta^{i+1} - \Delta^i}{\lambda_i^*} \quad \forall i \in [0, B-M-1],$$

*where for some $\lambda_i^* \in [\theta_{j-1}, \theta_j]$, $\Delta^i = C_j \cdot \lambda_i^* + D_j$.*

The proof of this theorem, as well as the complete pseudocode of the algorithm D-SETASIDE-GFQ, is provided in Appendix B.1. Additionally, in the special case of $K = 1$, OMcS with GFQ guarantee is closely related to the problem introduced [ZZZ15, JLTZ21] and with an extra assumption of $m_1 = 0$ it recovers the existing optimal result of [TYBLG25]. However, having multiple classes and GFQ constraints significantly increases the complexity of the problem.

## 3.2 Optimal Randomized Algorithm: R-SETASIDE-GFQ for OMcS with GFQ

We propose a randomized algorithm, termed Randomized Set-Aside with GFQ guarantee (R-SETASIDE-GFQ), and prove that it attains the optimal competitive ratio among all algorithms. R-SETASIDE-GFQ operates in two phases: in the first phase, the integral problem is relaxed to a fractional setting, and the optimal online decisions are computed in this relaxed space. In the second phase, these optimal fractional decisions are rounded to obtain a feasible integral solution. This approach is inspired by online correlated selection techniques originally developed in the online matching literature (e.g.,[HZZ20, FHTZ22]). Recently, [NSBT25] introduced a lossless online correlated $k$-rental rounding scheme that employs a single random seed to round fractional solutions in online selection problems with reusable resources, where items become available again after their rental periods expire. In contrast, for non-reusable settings, the rounding procedure follows Algorithm 2, in which a new random seed is independently sampled in each round to preserve the desired correlation structure across decisions. This random variable is drawn from a carefully designed Bernoulli distribution to ensure that each item's allocation remains synchronized across rounds, such that an item becomes available with the desired probability for allocation to an agent in each round. Building on this concept, our rounding scheme is designed to allocate items such that the expected performance of the integral solution mirrors that of the optimal fractional solution at every step. This ensures the algorithm maintains competitiveness and optimality at every step.

As previously mentioned, the optimal fractional decision at time $t$, denoted by $\tilde{x}_t \in [0, 1]$, is computed during the first relaxation phase and can be obtained using any optimal online fractional algorithm, denoted as FRAC-GFQ (line 8). An example of such an algorithm is provided in Algorithm 5 in Appendix B.3. In the subsequent stage, an integral solution $x_t$ is obtained using the lossless online

| **Algorithm 1:** Randomized Set-Aside with GFQ guarantee (R-SETASIDE-GFQ) | **Algorithm 2:** Lossless Online Rounding (ROUNDING) |
|---|---|

**Algorithm 1:** Randomized Set-Aside with GFQ guarantee (R-SETASIDE-GFQ)

**Input:** $B$; $\{m_j, \theta_j\}_{\forall j \in [K]}$
1 **Initialize:** Unit index $\kappa_1 = 1$ and $\{\kappa_1^j = 1\}_{\forall j \in [K]}$, utilization level $z_0 = 0$
2 **while** *agent $t$ arrives* **do**
3    Obtain agent $t$'s information $(v_t, \mathcal{J}_t)$
4    **if** $\kappa_t^j \leq m_j$ *for any $j \in \mathcal{J}_t$* **then**
5      $x_t = 1$
6      Update $\kappa_{t+1}^j = \kappa_t^j + x_t, \forall j \in \mathcal{J}_t$
7    **else**
8      $\tilde{x}_t = \text{FRAC-GFQ}(B, \{m_j, \theta_j\}_{\forall j})$
9      Update $z_t = z_{t-1} + \tilde{x}_t$
10      $x_t = \text{ROUNDING}(\kappa_t, z_t, z_{t-1}, \tilde{x}_t)$
11      Update $\kappa_{t+1} = \kappa_t + x_t$.

**Algorithm 2:** Lossless Online Rounding (ROUNDING)

**Input:** $\kappa, z_n, z_p, \tilde{x}$
1 **if** $\lceil z_n \rceil = \lceil z_p \rceil = \kappa$ **then**
2    $x = \begin{cases} 1 & \text{w.p. } \tilde{x}/(\lceil z_p \rceil - z_p) \\ 0 & \text{otherwise} \end{cases}$
3 **else if** $\lceil z_n \rceil \neq \lceil z_p \rceil$ **then**
4    **if** $\kappa = \lceil z_p \rceil$ **then**
5      $x = 1$    w.p. 1
6    **else if** $\kappa = \lceil z_n \rceil$ **then**
7      $x = \begin{cases} 1 & \text{w.p. } \frac{z_n - \lceil z_p \rceil}{(1 - \lceil z_p \rceil + z_p) \cdot (\lceil z_n \rceil - \lceil z_p \rceil)} \\ 0 & \text{otherwise} \end{cases}$

**Output:** $x$

rounding scheme of ROUNDING, given in Algorithm 2. This scheme ensures that the expected utility of the integral allocation aligns with the utility achieved in the fractional setting. Let $z_t = \sum_{t'=1}^{t} \tilde{x}_{t'}$ denote the fractional utilization level at time $t$. Specifically, if the fractional solution continues allocating item $\lceil z_{t-1} \rceil$, the rounding procedure allocates that item to the agent $t$ with probability $\tilde{x}_t/(\lceil z_{t-1} \rceil - z_{t-1})$, provided it is still available. On the other hand, if the fractional setting initiates the allocation of a new item, and the item $\lceil z_{t-1} \rceil$ in the integral solution remains available, it is allocated with probability 1. Otherwise, item $\lceil z_t \rceil$ is allocated probabilistically, maintaining the expectation of utility equivalence with the fractional solution. The following theorem states the main result regarding this multi-stage algorithm.

**Theorem 3.2** (OMcS with GFQ: Optimal Randomized Algorithm). *Given a* GFQ *requirement* **m**, *Algorithm 1 achieves the same competitive ratio as* FRAC-GFQ *for* OMcS *under the* GFQ *constraints, namely, the rounding scheme of Algorithm 2 is lossless.*

The proof of this theorem is presented in Appendix B.2. Note that when there is only 1 class and $m_1 = 0$, OMcS is reduced to the conventional online selection problem [LPS09], and Algorithm 1 achieves a tight competitive ratio $1 + \ln \theta_1$, which matches the lower bound [ZCL08, CZL08]. To the best of our knowledge, Algorithm 1 is the first randomized algorithm that can attain this result. Figure 1 shows the comparative ratio of R-SETASIDE-GFQ and D-SETASIDE-GFQ based on the number of available items (i.e., $B$). It illustrates that while these two ratios essentially converge for large values of $B$, R-SETASIDE-GFQ significantly outperforms D-SETASIDE-GFQ in cases with smaller inventory sizes. Furthermore, we believe that this rounding

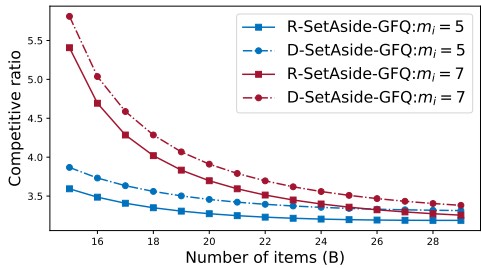

Figure 1: Comparison of D-SETASIDE-GFQ and R-SETASIDE-GFQ.

scheme can be adaptable to a wider range of related online selection problems, such as the single-leg revenue problem [BKK22], to achieve a tight guarantee. In the subsequent sections, we explore how this rounding scheme can be extended to address these additional problem settings, demonstrating its flexibility and effectiveness.

## 4 OMcS with $\beta$-Proportional Fairness

Despite its simplicity and intuitive appeal, the OMcS problem with GFQ guarantees exhibits several notable limitations. The reservation vector **m** is enforced as a hard constraint, yet determining appropriate values for **m** is often non-trivial and may be contentious in practice—e.g., whether to reserve an equal $1/K$ fraction of the total budget for each class or to allocate reservations

---

**Algorithm 3:** Randomized Set-Aside with $\beta$-PF guarantee (R-SETASIDE-PF)

---

**Input:** $B, \{\theta_j\}_{\forall j \in [K]}$.

**Initialize:** Sale unit index $\kappa_1 = 1$, utilization levels $\{z_0^{i,j} = 0\}_{\forall i,j \in [K]}$, $z_0^G = 0$ and $z_0 = 0$.

**1 while** *agent $t$ arrives* **do**

**2**    Obtain the value and class information of agent $t$: $v_t$ and $\mathcal{J}_t$ ;

**3**    **for** *all $i, j \in \mathcal{J}_t$* **do**                      ▷ Relaxation phase.

**4**       **if** $v_t \geq \phi_{i,j}(z_{t-1}^{i,j})$ **then**

**5**           $\hat{x}_t^{i,j} = \arg\max_{a \in [0,1]}\{av_t - \int_{z_{t-1}^{i,j}}^{z_{t-1}^{i,j}+a} \phi_{i,j}(\eta)d\eta\}$.

**6**    $\tilde{x}_t^{i,j} = \min\{[h - z_{t-1}^{i,j}]^+, \hat{x}_t^{i,j}\}$ and $h$ is the maximum number such that $\sum_{i,j \in \mathcal{J}_t} \tilde{x}_t^{i,j} \leq 1$.

**7**    **if** $v_t \geq \phi^G(u_{t-1})$ **then**

**8**       $\tilde{x}_t^G = \arg\max_{a \in [0, 1-\sum_{i,j \in \mathcal{J}_t} \tilde{x}_t^{i,j}]} \left\{a \cdot v_t - \int_{z_{t-1}^G}^{z_{t-1}^G+a} \phi^G(\eta)d\eta\right\}$.

**9**    Update $z_t^{i,j} = z_{t-1}^{i,j} + x_t^{i,j}$ for all $i, j \in \mathcal{J}_t$.

**10**    Update $z_t^G = z_{t-1}^G + x_t^G$.

**11**    Set $\tilde{x}_t = \sum_{i,j \in \mathcal{J}_t} \tilde{x}_t^{i,j} + \tilde{x}_t^G$ and $z_t = z_{t-1} + \tilde{x}_t$          ▷ Rounding phase.

**12**    $x_t = \text{ROUNDING}(\kappa_t, z_t, z_{t-1}, \tilde{x}_t)$;

**13**    Update $\kappa_{t+1} = \kappa_t + x_t$.

---

proportionally based on class sizes. Furthermore, as the algorithm must maintain feasibility without knowledge of future arrivals, early agents may receive disproportionately favorable allocations despite having low valuations, resulting in individual-level unfairness. To mitigate these challenges, we introduce a utility-based fairness notion that relaxes the rigidity of GFQ. Specifically, we propose Algorithm 3, which employs a relax-and-round framework [Ma24b]. In the following section, we first describe the relaxation phase (lines 3–10), which ensures $\beta$-proportional fairness under the multi-labeled setting. We then present a lossless rounding procedure (lines 11–13) that converts the fractional solution into a feasible integral allocation while preserving the fairness guarantees.

### 4.1 Relaxation Phase (lines 3-10): A Novel Fractional Set-Aside Algorithm

Here we first focus on the relaxation phase of Algorithm 3 where the allocation decisions can take fractional values. At a high level, this phase works as follows. Upon arrival of each agent, based on its class set and valuation information, $|\mathcal{J}_t| + \binom{|\mathcal{J}_t|}{2} + 1$ allocation decisions are made. The first $|\mathcal{J}_t|$ allocations are based on the agent's group-specific threshold function and the next $\binom{|\mathcal{J}_t|}{2}$ are based on the threshold functions designed for each pair of groups which can successfully ensures group fairness in the multi-labeled setting. The last one is based on a global threshold function, aimed at optimizing individual welfare. Specifically, we design $K + 1 + \binom{K}{2}$ threshold functions, one local threshold function for each class $j \in [K]$ denoted by $\phi_{j,j}(u) : [0, b_j] \to [1, \theta_j]$, one for each pair $i, j \in [K]$ denoted by $\phi_{i,j}(u) : [0, b_{ij}] \to [1, \min\{\theta_i, \theta_j\}]$, and one global threshold function, denoted by $\phi^G(u) : [0, B \cdot \mathfrak{b}] \to [1, \theta_K]$, where $\mathfrak{b} \in [0, 1]$ is a parameter indicating the importance of efficiency over fairness. In the following theorem we show that with a well-designed set of threshold functions, Algorithm 3 can smoothly balance efficiency and fairness.

**Theorem 4.1.** *For any given $\mathfrak{b} \in [0, 1]$, the relaxation phase of Algorithm 3 is $\alpha(\mathfrak{b})$-competitive (in utility maximization) and $\beta(\mathfrak{b})$-PF, where*

$$\alpha(\mathfrak{b}) = \frac{1}{\frac{1-\mathfrak{b}}{\sum_{j \in [K]}(K-j+1)\cdot\alpha_j} + \frac{\mathfrak{b}}{\alpha_K}}, \qquad \beta(\mathfrak{b}) = \frac{1}{1-\mathfrak{b}} \cdot \sum_{j \in [K]} \left(1 - \frac{j-1}{K}\right) \cdot \alpha_j, \qquad (2)$$

*provided that for all $i \geq j \in [K]$, the threshold functions are designed as follows:*

$$\phi_{j,i}(u) = \begin{cases} 1 & u \in \left[0, \frac{b_j}{\alpha_j}\right], \\ \exp\left(\frac{K \cdot \bar{\beta}(\mathfrak{b}) \cdot u}{B} - 1\right) & u \in \left[\frac{b_j}{\alpha_j}, b_j\right], \end{cases} \quad \phi^G(u) = \begin{cases} 1 & u \in \left[0, \frac{B \cdot \mathfrak{b}}{\alpha_K}\right], \\ \exp\left(\frac{\alpha_K \cdot u}{B \cdot \mathfrak{b}} - 1\right) & u \in \left[\frac{B \cdot \mathfrak{b}}{\alpha_K}, B \cdot \mathfrak{b}\right], \end{cases}$$

*where $b_j = \frac{B \cdot \alpha_j \cdot (1-\mathfrak{b})}{\sum_{i \in [K]} (K-i+1) \cdot \alpha_i}$ with $\alpha_i = 1 + \ln \theta_i$.*

The proof of Theorem 4.1 is given in Appendix C.1. Here, the parameter $\mathfrak{b} \in [0,1]$ quantifies the degree of emphasis placed on efficiency. Notably, as $\mathfrak{b} \to 1$, the allocation is governed solely by the global threshold function and ignore the set-aside budget to guarantee fairness, causing the algorithm to converge to the optimal competitive ratio $\alpha_K$ without fairness constraints. On the other hand, as $\mathfrak{b} \to 0$, the algorithm excludes the global threshold function entirely and split the set-aside budget among class-based threshold function, which achieves $\left( \frac{1}{K} \sum_{i \in [K]} (K-i+1) \cdot \alpha_i \right)$-PF.

We note that [BGH+23] studied a generalized version of this problem, where valuations are not uniform across groups at each timestep. However, our approach significantly diverges from theirs. Their algorithm reserves half of the total budget upfront to ensure fairness and greedily allocates the remainder to minimally satisfy $\beta$-proportional fairness at each step. In contrast, our method employs carefully designed threshold functions that provide a more principled and flexible mechanism for allocation. Moreover, their design makes it difficult to explore the trade-off between fairness and efficiency, whereas our approach allows for a more transparent and systematic examination of this relationship. We further note that in the special case where $|\mathcal{J}_t| = 1$ for all $t \in [T]$–that is, when there are no multi-labeled arrivals–the need for reserving budgets for each class pair disappears. In this setting, our results recover those of [ZJST25], which proposed an algorithm that achieves the Pareto-optimal trade-off between fairness and efficiency. The following corollary formally states this result.

**Corollary 4.2** (Pareto-optimality). *For fractional $\mathsf{OMcS}$ with single-labeled agents only, Algorithm 3 is Pareto-optimal in that for any $\mathfrak{b} \in [0,1]$ and $\epsilon > 0$, no online algorithm can be $(\alpha(\mathfrak{b}) - \epsilon)$-competitive without deteriorating the fairness guarantee (i.e., increasing the value of $\beta(\mathfrak{b})$).*

The proof of Corollary 4.2 is provided in Appendix C.2. While this result establishes the optimal fairness-efficiency trade-off in the single-labeled setting, the same does not hold for the algorithm in [ZJST25] when extended to the multi-labeled case. Specifically, under multi-labeled arrivals, their method guarantees only a $\left( \frac{1}{1-\mathfrak{b}} \cdot \sum_{j \in [K]} \alpha_j \right)$-PF solution, which is significantly weaker than the fairness guarantee achieved by our proposed algorithm. This corollary therefore not only confirms the Pareto-optimality of our design in the single-labeled regime but also illustrates its superior performance and generalization to the multi-labeled setting in $\mathsf{OMcS}$.

## 4.2 Rounding Phase (lines 11-13): A Lossless Online Rounding Scheme

In this section, we discuss the rounding phase of Algorithm 3, which builds upon the rounding scheme described in Algorithm 2. This algorithm leverages the concept of negative correlation among decisions to achieve the optimal $\beta$-proportional fairness guarantee. In particular, at each time $t$, Algorithm 3 first computes at most $|\mathcal{J}_t| + \binom{|\mathcal{J}_t|}{2} + 1$ provisional allocations based on the class-specific threshold functions to enforce fairness, and using a global threshold function to promote overall efficiency. The sum of these allocations yields a total fractional allocation at time $t$, which is then rounded to an integral allocation via the rounding scheme of Algorithm 2. Somewhat surprisingly, the rounding scheme of Algorithm 2, originally developed for $\mathsf{OMcS}$ with GFQ constraints and proven to be lossless in that setting, also serves as a lossless online rounding scheme for $\mathsf{OMcS}$ under the $\beta$-PF guarantee. Specifically, it preserves the performance guarantee obtained in the relaxation phase when transitioning to the integral setting. Theorem 4.3 below highlights this in detail.

**Theorem 4.3** ($\mathsf{OMcS}$ with $\beta$-PF). *For any $\mathfrak{b} \in [0,1]$, Algorithm 3 is $\alpha(\mathfrak{b})$-competitive and $\beta(\mathfrak{b})$-PF, where $\alpha(\mathfrak{b})$ and $\beta(\mathfrak{b})$ are defined in Eq.* (2).

The proof of this theorem is presented in Appendix C.3. Since based on Corollary 4.2, Algorithm 3 recovers the Pareto-optimal design of [ZJST25] in the special case of single-labeled setting where $|\mathcal{J}_t| = 1$ for all $t \in [T]$, the final integral allocation is also Pareto-optimal because the rounding phase does not deviate the performance guarantee.

Before concluding this section, we briefly comment on the computational complexity of the proposed algorithms. All algorithms operate in an online manner and require only $\mathcal{O}(1)$ time per buyer arrival. For instance, Algorithm 3 computes the fractional allocation $\tilde{x}_t$ by solving a convex pseudo-revenue maximization problem (lines 5 and 8), using predefined threshold functions for each class. The resulting fractional allocation is then rounded in $\mathcal{O}(1)$ time to yield the final decision (line 12).

# 5 Improving Group Fairness via Learning-Augmented Algorithms

The $\beta$-proportional fairness essentially ensures that the desired allocation $\mathbf{x}$ is comparable to all other allocations $\mathbf{w}$ in the PF sense, i.e., $\frac{1}{K} \sum_{j \in [K]} \frac{U_j(\mathbf{w})}{U_j(\mathbf{x})} \leq \beta$. However, this can be overly pessimistic in practice. To address this, predictions from machine learning tools or advice from experts about a fair allocation are often available, and can be used to improve fairness guarantees. In this section, we aim to explore how to leverage (possibly imperfect) advice about a fair allocation based on the *consistency-robustness* framework in the literature of learning-augments algorithms [WZ20, KPS18].

Specifically, an allocation $\mathbf{x}$ is called $\eta$-***consistent proportional fair*** if it satisfies $\eta$-proportional fairness with respect to the advice allocation $\hat{\mathbf{x}}$, i.e., $\frac{1}{K} \sum_{j \in [K]} \frac{U_j(\hat{\mathbf{x}})}{U_j(\mathbf{x})} \leq \eta$. Similarly, an allocation $\mathbf{x}$ is $\gamma$-***robust proportional fair*** if it satisfies $\gamma$-proportional fairness with respect to any allocation $\mathbf{w}$, $\frac{1}{K} \sum_{j \in [K]} \frac{U_j(\mathbf{w})}{U_j(\mathbf{x})} \leq \gamma$. These metrics allow us to balance the benefits of good advice with the need for robustness against advice errors, ensuring a more practical and reliable fairness guarantee.

**Definition 3** (Advice Model of OMcS with $\beta$-PF). For the OMcS problem with $\beta$-PF guarantee, we define $\mathsf{ADV} := \hat{\mathbf{x}} = \{\hat{x}_t \in \{0, 1\} : t \in [T]\}$ as the untrusted fair advice.

Note that the primary objective of this section is to use this advice to improve the fairness guarantee. Therefore it can not recover the 1-consistency of the efficiency even when the prediction is completely correct. Moreover, since the focus of this advice is on improving fairness, we set $\mathfrak{b} = 0$ throughout, thereby placing full emphasis on the fairness objective. We introduce a learning-augmented algorithm, termed the *Linear Combination-based Learning-Augmented Algorithm* (LiLA), for the OMcS problem with a $\beta$-PF guarantee. At each time step $t$, the algorithm generates two candidate decisions: a *robust decision* $\bar{x}_t$, computed using Algorithm 3, and a *predicted fair decision* $\hat{x}_t$, produced by a black-box machine learning model trained on data from ADV. The algorithm then selects its final decision $x_t$ through a randomized mechanism that depends on the level of trust in the prediction. A hyperparameter $\epsilon$ quantifies the confidence in the prediction and serves as a control variable balancing *consistency* and *robustness*: as $\epsilon \to 0$, the algorithm becomes more consistent with the predictions but less robust to errors. Accordingly, a combination probability $\rho \in [0, 1]$, determined by $\epsilon$, regulates the decision-maker's reliance on the black-box advice. Specifically, $\rho$ denotes the probability of adopting the predicted decision $\hat{x}_t$, while $1 - \rho$ denotes the probability of following the robust decision $\bar{x}_t$. Hence, the expected decision of the learning-augmented algorithm at each time step is $x_t = \rho \hat{x}_t + (1 - \rho)\bar{x}_t$.

For a hyperparameter $\epsilon \in [0, \beta - 1]$, where $\beta = \beta(0)$ represents the proportional fairness guarantee of Algorithm 3, $\rho$ is defined as $\rho := \left(\frac{\beta}{1+\epsilon} - 1\right) \cdot \frac{1}{\beta - 1}$. It is easy to verify that $\rho \in [0, 1]$. When $\epsilon = 0$, the algorithm fully trusts the advice decisions, setting $\rho = 1$. On the other hand, when $\epsilon = \beta - 1$, the algorithm reverts entirely to the robust algorithm by setting $\rho = 0$. The following theorem presents our main results regarding the consistency and robustness of LiLA.

**Theorem 5.1.** *For any $\epsilon \in [0, \beta - 1]$,* LiLA *for* OMcS *with $\beta$-PF guarantee is $(1 + \epsilon)$-consistent proportional fair and $\frac{(1+\epsilon)(\beta-1)}{\epsilon}$-robust proportional fair.*

The proof of the above theorem is provided in Appendix D.1. It is observed that as $\epsilon \to 0$, the algorithm becomes 1-consistent, but its robustness is unbounded. This behavior is intuitive: when the algorithm fully relies on the untrusted advice, there is a risk that the advice is inaccurate, leading to an unbounded fairness guarantee.

Furthermore, we can see that in the single-labeled arrival setting, Theorem 5.1 implies Pareto optimality of consistency-robustness trade-off. This is details in Appendix D.2. From our analysis in this special case, LiLA reserves $b_j = \frac{B}{K \cdot \eta} + \frac{B \cdot \ln \theta_j}{K \cdot \gamma}$ for each class $j \in [K]$, where $\eta$ and $\gamma$ represent the consistency and robustness parameters. Notably, when the advice is accurate (i.e., $\epsilon \to 0$), each class receives exactly $\frac{1}{K}$ of the resource. This matches the optimal reservation achieved by NSW, as presented in [BGH$^+$23], since 1-PF is equivalent to 1-NSW.

The following corollary summarizes the main results regarding the efficiency of this model:

**Corollary 5.2.** *For any $\epsilon \in [0, \beta-1]$,* LiLA *for* OMcS *with $\beta$-PF guarantee is $(K \cdot (1+\epsilon))$-consistent competitive and $\frac{K \cdot (1+\epsilon)(\beta-1)}{\epsilon}$-robust competitive.*

The above corollary holds since any $\beta$-PF allocation $\mathbf{x}$ is guaranteed to be $(K\beta)$-competitive, as for any allocation $\mathbf{w}$, we can see that $\frac{\sum_{j \in [K]} U_j(\mathbf{w})}{\sum_{j \in [K]} U_j(\mathbf{x})} \leq \sum_{j \in [K]} \frac{U_j(\mathbf{w})}{U_j(\mathbf{x})} \leq K\beta$. Intuitively, achieving

1-consistency in fairness necessitates reserving $\frac{1}{K}$ of the resources for each class. This leads to a reduction in efficiency by a factor of $K$ when all arrivals belong to a single class, as the efficient algorithm would allocate the entire resource to that class. Consequently, even with perfectly accurate predictions, the algorithm ensures $K$-consistent efficiency. This trade-off arises because the algorithm prioritizes fairness over efficiency in its design. We also demonstrated in Appendix D.4 that LiLA can also enhance the performance guarantee of the OMcS problem with GFQ guarantee. The key distinction lies in the advice model, which is assumed to always satisfy GFQ requirement, while the prediction focuses on improving efficiency.

## 6   Conclusion and Future Work

In this paper, we studied group fairness guarantees in the online multi-class selection problem under a multi-labeled agent setting. We proposed a novel randomized algorithm based on a relax-and-round framework, where a carefully designed rounding step ensures that the integral solution matches the performance of the fractional one in expectation—addressing a key limitation of existing methods. Our algorithm is specifically designed to handle the complexities introduced by multi-labeled agents, enabling fair and efficient allocation across overlapping classes. To further improve performance beyond worst-case guarantees, we also developed a learning-augmented variant that incorporates untrusted predictions to enhance average-case outcomes.

Our work opens several intriguing directions for future research. A key question is how our results extend to alternative arrival models, such as the random order model or stochastic i.i.d. settings, and what implications these extensions may have for online fair allocation in broader contexts such as mechanism design, auctions, and multi-agent systems. Another important direction is to investigate whether our findings can be generalized to multi-resource settings (e.g., combinatorial auctions), in both fractional and integral forms.

## Acknowledgments

Xiaoqi Tan acknowledges support from Alberta Machine Intelligence Institute (Amii), Alberta Major Innovation Fund, and NSERC Discovery Grant RGPIN-2022-03646.

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

# A Numerical Experiments

In this section, we evaluate our algorithms to investigate the empirical fairness and efficiency under different fairness notions and algorithms. We also explore how untrusted black-box advice can help the improvement of fairness.

## A.1 Experimental Setup

We evaluate our theoretical results using the Google Cluster Data [Goo15], which records CPU usage over time for three request types and here we consider them as distinct classes. The normalized CPU allocations are scaled to fit our model, and requests needing more than one CPU units are split into single-unit requests. We assign valuations randomly with $\theta_1 = 5$, $\theta_2 = 10$, and $\theta_3 = 15$, with $B = 100$. We analyze OMcS with GFQ under deterministic (D-SETASIDE-GFQ) and randomized (R-SETASIDE-GFQ) algorithms, denoted as d-GFQ and r-GFQ respectively. For our analysis, we set $m_j = 5$ for each class $j \in [K]$. We also study OMcS with $\beta$-proportional fairness denoted as $\beta$-PF (R-SETASIDE-PF with $\mathfrak{b} = 1$) and without fairness consideration denoted as $\alpha$-CR (R-SETASIDE-PF with $\mathfrak{b} = 1$). To model the advice, we follow [LCS$^+$24] approach. Let $\xi \in [0, 1]$ represents an *adversarial probability*. When $\xi = 0$, ADV provides the optimal solution, and when $\xi = 1$, ADV is fully adversarial. Formally, let $\{x_t^* : t \in [T]\}$ denote the optimal decisions and $\{\check{x}_t : t \in [T]\}$ the decisions that minimize the objective. Then, in expectation, the advised decisions are given by ADV $= \{(1 - \xi)x_t^* + \xi\check{x}_t : t \in [T]\}$. Under this advice model, we examine LiLA with GFQ (GFQ-LA) and $\beta$-proportional fairness ($\beta$-PF-LA) as well.

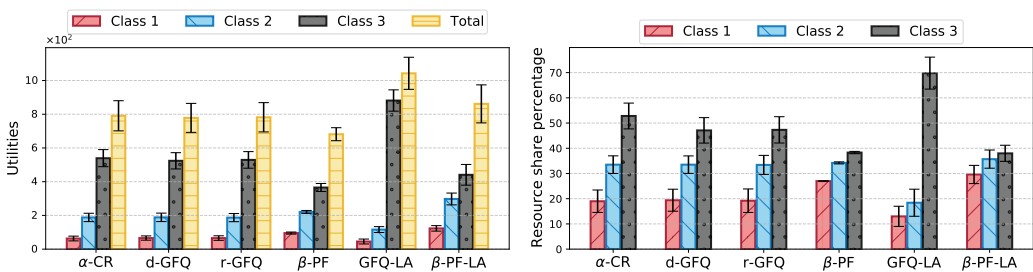

Figure 2: Utilities and resource allocations of each class under different algorithms; $\theta_1 = 5$, $\theta_2 = 10$ and $\theta_3 = 15$.

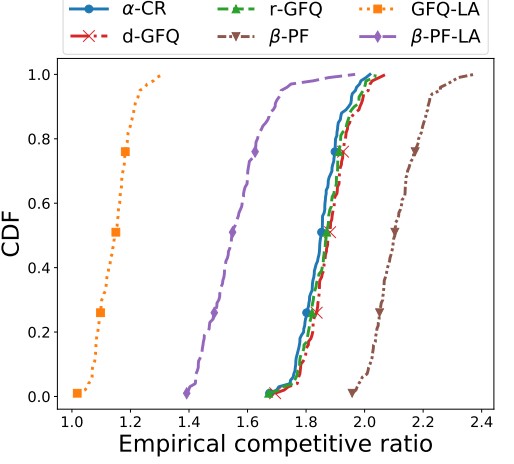

Figure 3: CDF of empirical competitive ratios of different algorithms.

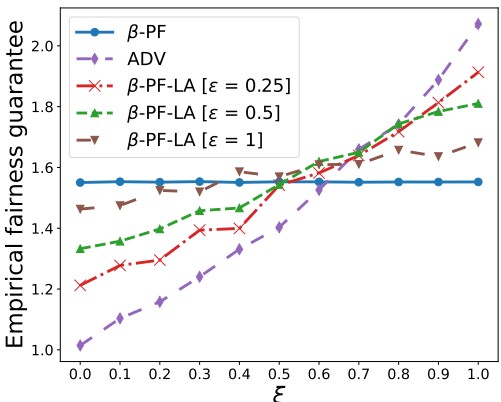

Figure 4: Variation of the empirical fairness guarantee w.r.t. the adversarial probability $\xi$.

## A.2 Experimental Results

In our first experiment, we set $\xi = 0$ and $\epsilon = 1.25$. Figure 2 illustrates the utilities achieved by each class and the corresponding resource allocations. As observed, the $\beta$-PF algorithms allocate resources more equitably across classes while maintaining a high level of efficiency in terms of total utility. In contrast, quantity-based approaches prioritize efficiency over fairness, resulting in higher total utility but less equitable allocation. Furthermore, the superiority of the LiLA algorithms is evident in this figure when compared to those without predictions. Furthermore, Figure 3 presents the cumulative distribution function (CDF) of the empirical competitive ratio. This figure demonstrates how predictions enhance performance by improving the convergence of the competitive ratio. Additionally, it highlights the superiority of r-GFQ compared to d-GFQ.

In another experiment, we investigate the impact of prediction accuracy and the reliance of algorithms on these predictions on the empirical $\beta$-PF guarantee. We consider $\beta$-PF-LA with different values of $\epsilon$, corresponding to varying combination probabilities $\rho$, along with the algorithm without predictions and the algorithm fully relying on predictions (ADV). Figure 4 illustrates that greater reliance on predictions improves the fairness guarantee when the advice is of high quality but worsens it as the prediction quality decreases. This highlights the trade-off between consistency and robustness.

# B    Section 3 Proofs

## B.1    Proof of Theorem 3.1

We introduce a deterministic threshold-based online algorithm, D-SETASIDE-GFQ, that is optimal among all deterministic algorithms. D-SETASIDE-GFQ relies on a set of thresholds, denoted as $\boldsymbol{\lambda} = \{\lambda_i\}_{i \in [B-M]}$, which are used to decide whether to accept or reject an arriving item. The objective is to design these thresholds in a way that minimizes the competitive ratio of the algorithm.

---

**Algorithm 4:** Deterministic Set-Aside with GFQ guarantee (D-SETASIDE-GFQ)

**Input:** $B$; $\{m_j\}_{\forall j \in [K]}$; $\{\lambda_i^*\}_{\forall i \in [B-M]}$.

**Initialize:** Unit index $\kappa_1 = 1$, $\{\kappa_1^j = 1\}_{\forall j \in [K]}$.

1 **while** *buyer $t$ arrives* **do**
2      Obtain agent $t$'s information $(v_t, \mathcal{J}_t)$
3      **if** $\kappa_t^j \leq m_j$ *for any $j \in \mathcal{J}_t$* **then**
4          $x_t = 1$
5          Update $\kappa_{t+1}^j = \kappa_t^j + x_t, \forall\, j \in \mathcal{J}_t$
6      **else**
7          Decide the selection according to:
8          **if** $\kappa_t \leq B - M$ **and** $v_t \geq \lambda^*_{\kappa_t - 1}$ **then**
9              $x_t = 1$
10              Update $\kappa_{t+1} = \kappa_t + x_t$

---

Here, we first prove that Algorithm 4, with the thresholds designed in Theorem 3.1, always achieves $\alpha^*$-competitiveness. We then prove the optimality of this design compared to any other deterministic algorithm.

For a fixed input sequence $I$, let the algorithm terminate after allocating $Z$ out of $B$ units of resources, obtaining a value of $\mathsf{ALG}(I)$. Let $S$ and $S'$ be the sets of items selected by Algorithm 4 and the optimal solution, respectively. We denote the number and value of the common items by $W = |S \cap S'|$ and $V = \sum_{t \in S \cap S'} v_t$. Since the admission thresholds are monotonically increasing, we observe that for any item $j$ not selected by the algorithm, $v_j \leq \lambda_Z^*$. As a result,

$$\mathsf{OPT}(I) \leq V + \lambda_{Z-M}^* \cdot (B - W).$$

Let $V' = \sum_{t \in (S/S')} v_t$ be the value of those items that are selected by the algorithm and not selected by the offline optimum. As a result, we can see that:

$$\frac{\mathsf{OPT}(I)}{\mathsf{ALG}(I)} \leq \frac{V + \lambda_{Z-M}^* \cdot (B - W)}{V + V'}.$$

Since each item $j$ picked in $S$ after the satisfying the GFQ constraint and selected as the $i$-th item must have valuation at least $\lambda_{i-1}^*$, we have:

$$V \geq \sum_{t \in S \cap S'} \lambda_t^* + M_1, \quad \text{call this } V_1,$$

$$V' \geq \sum_{tS/S'} \lambda_t^* + M_2, \quad \text{call this } V_2,$$

where $M_1$ and $M_2$ are the part of GFQ constraints that are satisfied in $V$ and $V'$, respectively and $M_1 + M_2 = M$. Since $\mathsf{OPT}(I) \geq \mathsf{ALG}(I)$, we can see that:

$$\frac{\mathsf{OPT}(I)}{\mathsf{ALG}(I)} \leq \frac{V + \lambda_{Z-M}^* \cdot (B - W)}{V + V'} \leq \frac{V_1 + \lambda_{Z-M}^* \cdot (B - W)}{V_1 + V'} \leq \frac{V_1 + \lambda_{Z-M}^* \cdot (B - W)}{V_1 + V_2}.$$

Additionally, by monotonicity of admission thresholds, we get $V_1 \leq \lambda_{Z-M}^*(W - \max_{i \in [j-1]}\{m_i\}) + \sum_{i=1}^{j-1}[m_i - \max_{l \in [i-1]}\{m_l\}]^+ \cdot \theta_i$ when $\lambda_{Z-M}^*$ is in $[\theta_{j-1}, \theta_j]$. Furthermore, $V_1 + V_2 = M + \sum_{i=0}^{Z-M-1} \lambda_i^*$. As a result we get

$$\frac{\mathsf{OPT}(I)}{\mathsf{ALG}(I)} \leq \frac{\lambda_{Z-M}^* \cdot (B - \max_{i \in [j-1]}\{m_i\}) + \sum_{i=1}^{j-1}[m_i - \max_{l \in [i-1]}\{m_l\}]^+ \cdot \theta_i}{M + \sum_{i=0}^{Z-M-1} \lambda_i^*}$$

$$\leq \frac{\lambda_{Z-M}^* \cdot C_j + D_j}{M + \sum_{i=0}^{Z-M-1} \lambda_i^*}.$$

Based on the system of equations presented in Theorem 3.1, it is easy to verify that $\frac{\lambda_{Z-M}^* \cdot C_j + D_j}{M + \sum_{i=0}^{Z-M-1} \lambda_i^*} = \alpha^*$ and this concludes the $\alpha^*$-competitiveness of Algorithm 4.

Now, we will discuss the optimality of the presented design in Theorem 3.1. Let us first introduce a hard instance for the OMcS problem with GFQ constraints.

**Definition B.1** (GFQ Fairness Guarantee Hard Instance in Integral Setting: $\mathcal{I}^{i-\mathsf{GFQ}}$). Instance $I^{i-\mathsf{GFQ}}$ is defined as a scenario characterized by a continuous, non-decreasing sequence of valuation arrivals. In this scenario, each valuation is replicated for every class as long as it remains feasible and a copy of this valuation which has the label of all possible classes. For some value of $\epsilon$ such that $\epsilon \to 0$, instance $I^{i-\mathsf{GFQ}}$ can be shown as follows:

$$I^{i-\mathsf{GFQ}} = \left\{ \underbrace{(1, \{1\}), (1, \{2\}), \dots, (1, \{K\}), (1, \{1, \dots, K\})}_{K+1 \text{ number of agents}}, \right.$$

$$\underbrace{(1 + \epsilon, \{1\}), \dots, (1 + \epsilon, \{K\}), (1 + \epsilon, \{1, \dots, K\})}_{K+1 \text{ number of agents}}, \dots,$$

$$\underbrace{(\theta_1, \{1\}), \dots, (\theta_1, \{K\}), (\theta_1, \{1, \dots, K\})}_{K+1 \text{ number of agents}},$$

$$\left. \underbrace{(\theta_1 + \epsilon, \{2\}), \dots, (\theta_1 + \epsilon, \{K\}), (\theta_1 + \epsilon, \{2, \dots, K\})}_{K \text{ number of agents}}, \dots, \underbrace{(\theta_k, \{K\})}_{1 \text{ agent}} \right\},$$

where in above $(v, \mathcal{J})$, $\forall j \in [k]$, corresponds to the $B$ copies of an agent with valuation equal to $v$ with the label set $\mathcal{J}$.

Now, under this instance, any deterministic algorithm should reserve $M$ items to ensure that the GFQ constraint is always satisfied. Additionally, it should select $B - M + 1$ admission prices, denoted by $\boldsymbol{\lambda}$, in a way that minimizes the competitive ratio at any stopping point of the sequence $I^{i-\mathsf{GFQ}}$. When the stopping point is 1, the optimal offline solution will allocate the entire capacity at this price. However, the online algorithm cannot do such things since the adversary might send much higher prices followed by this stream and penalize the algorithm. Therefore, in order to maintain the $\alpha$-competitiveness, the algorithm will allocate only $\tau$ units to the agents with valuation 1 where $\tau$ is in a way that:

$$M + \tau + 1 \geq \max\left\{M, \frac{B}{\alpha^*}\right\}.$$

In this way, it is always possible to guarantee the $\alpha$-competitiveness when all the valuations are 1.

Now we consider two cases based on the value of $M$. By starting the stream of higher prices, the algorithm will also set the higher admission thresholds in a way the following equation system satisfies.

**Case 1.** As the first case, we consider the situation where $M \leq B/\alpha$. For any arbitrary small $\delta$, the system of equations is

$$\begin{cases} M + (\tau + 1) \cdot 1 = \frac{1}{\alpha} \left[ C_1 \cdot (\lambda_{\tau+1} - \delta) + D_1 \right] = \frac{1}{\alpha} \cdot \Delta^{\tau+1}, \\ M + (\tau + 1) \cdot 1 + \lambda_{\tau+1} = \frac{1}{\alpha} \left[ C_1 \cdot (\lambda_{\tau+2} - \delta) + D_1 \right] = \frac{1}{\alpha} \cdot \Delta^{\tau+2}, \\ \vdots \\ M + (\tau + 1) \cdot 1 + \lambda_{\tau+1} + \cdots + \lambda_{B-M-1} = \frac{1}{\alpha} \left[ C_K \cdot (\lambda_{B-M} - \delta) + D_K \right] = \frac{1}{\alpha} \cdot \Delta^{B-M}, \end{cases}$$

which implies that

$$\alpha = \frac{\Delta^{\tau+1}}{\tau + 1} = \frac{\Delta^{i+1} - \Delta^i}{\lambda_i} \quad \forall i \in [\tau + 1, B - M - 1],$$

where $\lambda_i \in [\theta_{j-1}, \theta_j]$ and $\Delta^i = Cj \cdot \lambda_i + D_j$.

**Case 2.** In the second case, we consider the situation where $M > B/\alpha$. In this case there is no need to set any admission threshold at 1 since by maintaining the GFQ constraint, the $\alpha$-competitiveness is guaranteed automatically when all the valuations is at 1. As a result, for any arbitrary small $\delta$, the system of equations is

$$\begin{cases} M \cdot 1 = \frac{1}{\alpha} \left[ C_{j^*} \cdot (\lambda_0 - \delta) + D_{j^*} \right] = \frac{1}{\alpha} \cdot \Delta^0, \\ M \cdot 1 + \lambda_0 = \frac{1}{\alpha} \left[ C_{j^*} \cdot (\lambda_1 - \delta) + D_{j^*} \right] = \frac{1}{\alpha} \cdot \Delta^1, \\ \vdots \\ M \cdot 1 + \lambda_0 + \cdots + \lambda_{B-M-1} = \frac{1}{\alpha} \left[ C_K \cdot (\lambda_{B-M} - \delta) + D_K \right] = \frac{1}{\alpha} \cdot \Delta^{B-M}, \end{cases}$$

where $j^*$ is a class index such that $M = \frac{1}{\alpha}(C_{j^*}\lambda_0 + D_{j^*})$ and $\lambda_0 \in [\theta_{j^*-1}, \theta_{j^*}]$. This implies

$$\alpha = \frac{\Delta^{\tau+1}}{\tau + 1} = \frac{\Delta^{i+1} - \Delta^i}{\lambda_i} \quad \forall i \in [\tau + 1, B - M - 1],$$

where $\lambda_i \in [\theta_{j-1}, \theta_j]$ and $\Delta^i = C_j \cdot \lambda_i + D_j$. We thus conclude the optimality of the design in Theorem 3.1.

Figure 5 shows the thresholds in different settings of OMcS with various GFQ constraints, denoted as d-GFQ. These are compared to the OMcS without fairness consideration, based on the algorithm designed in [TYBLG25], denoted as d-DYNAMIC. As pointed out by prior studies such as [TYBLG25], this algorithm is simple and fails matching the lower-bound specially in low inventory cases.

### B.2 Proof of Theorem 3.2

We use the mathematical induction to prove that the expected performance of Algorithm 1 is equal to the fractional performance at every time steps. Let $\tilde{x}_t$ be the decision of a fractional algorithm FRAC-GFQ and $U^t = \sum_{t'=0}^{t} v_{t'} \cdot \tilde{x}_{t'}$ be the cumulative utility of the agents up to time $t$ from the fractional solution. Now let us consider the base case and let $\bar{t}$ be the first time step after satisfying GFQ constraints such that $\tilde{x}_{\bar{t}} \in (0, 1]$. This means that, at this time, the optimal fractional algorithm allocates an item fractionally to the $\bar{t}$-th buyer. Thus in this case, $\kappa_{\bar{t}} = \lceil z_{\bar{t}} \rceil$. As a result, the expected utility of Algorithm 1 is

$$\sum_{t=1}^{\bar{t}} \mathbb{E}[v_t \cdot x_t] = v_{\bar{t}} \cdot \mathbb{E}[x_{\bar{t}}] = v_{\bar{t}} \cdot (1 \cdot (z_{\bar{t}} - \lceil z_{\bar{t}-1} \rceil)) = v_{\bar{t}} \cdot (1 \cdot (z_{\bar{t}} - z_{\bar{t}-1})) = v_{\bar{t}} \cdot \tilde{x}_{\bar{t}} = U^{\bar{t}}.$$

Now it is sufficient to show the induction step to complete this proof. Let assume the expected performance of Algorithm 1 up to time $\hat{t}$ satisfies $\sum_{t=1}^{\hat{t}} \mathbb{E}[v_t \cdot x_t] = U^{\hat{t}}$. Now we need to prove that $\sum_{t=1}^{\hat{t}+1} \mathbb{E}[v_t \cdot x_t] = U^{\hat{t}+1}$. The availability probability of item $\lceil z_{\hat{t}} \rceil$ at time $\hat{t}$ is $\lceil z_{\hat{t}} \rceil - z_{\hat{t}}$, which is proven based on induction as follows. Let us write the availability probability of this item as:

$$P(\text{item } \lceil z_{\hat{t}} \rceil \text{ available}) = P(\text{item } \lceil z_{\hat{t}} \rceil \text{ available}|\text{item } \lceil z_{\hat{t}-1} \rceil \text{ available}) \cdot P(\text{item } \lceil z_{\hat{t}-1} \rceil \text{ available}) +$$
$$P(\text{item } \lceil z_{\hat{t}} \rceil \text{ available}|\text{item } \lceil z_{\hat{t}-1} \rceil \text{ not available}) \cdot P(\text{item } \lceil z_{\hat{t}-1} \rceil \text{ not available}).$$

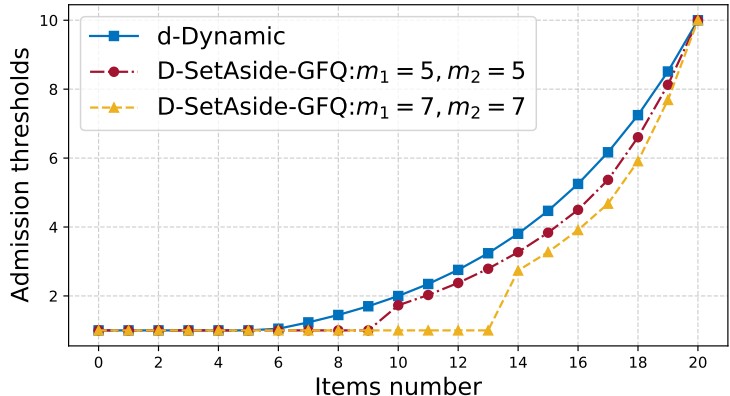

Figure 5: The red and yellow curves (D-SETASIDE-GFQ) show the impact of GFQ constraint compare to the blue curve which shows the thresholds without any fairness consideration (d-DYNAMIC). This figure shows that higher total GFQ requirements leads to having lower admission thresholds. Here we set $B = 20$, $\theta_1 = 5$ and $\theta_2 = 10$.

Now in the case $\lceil z_{\hat{t}} \rceil = \lceil z_{\hat{t}-1} \rceil$, the probability of item $\lceil z_{\hat{t}-1} \rceil$ being available is 0. As a result, we have

$$P(\text{item } \lceil z_{\hat{t}} \rceil \text{ available}) = \left(1 - \frac{\tilde{x}_{\hat{t}-1}}{\lceil z_{\hat{t}-1} \rceil - z_{\hat{t}-1}}\right) \cdot (\lceil z_{\hat{t}-1} \rceil - z_{\hat{t}-1}) + 0$$

$$= \lceil z_{\hat{t}-1} \rceil - z_{\hat{t}-1} - \tilde{x}_{\hat{t}-1} = \lceil z_{\hat{t}} \rceil - z_{\hat{t}}.$$

Now when $\lceil z_{\hat{t}} \rceil \neq \lceil z_{\hat{t}-1} \rceil$ we have

$$P(\text{item } \lceil z_{\hat{t}} \rceil \text{ available}) = 1 \cdot (\lceil z_{\hat{t}-1} \rceil - z_{\hat{t}-1}) + \left(1 - \frac{z_{\hat{t}} - \lceil z_{\hat{t}-1} \rceil}{1 - \lceil z_{\hat{t}-1} \rceil + z_{\hat{t}-1}}\right) \cdot (1 - \lceil z_{\hat{t}-1} \rceil + z_{\hat{t}-1})$$

$$= \lceil z_{\hat{t}-1} \rceil - z_{\hat{t}-1} + 1 + z_{\hat{t}-1} - z_{\hat{t}} = \lceil z_{\hat{t}-1} \rceil + 1 - z_{\hat{t}} = \lceil z_{\hat{t}} \rceil - z_{\hat{t}}.$$

This concludes that at each time $\hat{t}$, item $\lceil z_{\hat{t}} \rceil$ is available with probability $\lceil z_{\hat{t}} \rceil - z_{\hat{t}}$. Additionally, since the utilities are linear we have $\sum_{t=1}^{\hat{t}+1} \mathbb{E}[v_t \cdot x_t] = U^{\hat{t}} + v_{\hat{t}+1} \cdot \mathbb{E}[x_{\bar{t}+1}]$.

Based on the algorithm, there are two possible cases regarding the allocated item at time $\hat{t} + 1$:

**Case 1 -** $\lceil z_{\hat{t}+1} \rceil = \lceil z_{\hat{t}} \rceil$**:** In this case the only possibility is to allocate item $\lceil z_{\hat{t}} \rceil$. This item is available with probability $\lceil z_{\hat{t}} \rceil - z_{\hat{t}}$. Therefore, the expected utility in this case is:

$$\sum_{t=1}^{\hat{t}+1} \mathbb{E}[v_t \cdot x_t] = U^{\hat{t}} + v_{\hat{t}+1} \cdot \mathbb{E}[x_{\bar{t}+1}]$$

$$= U^{\hat{t}} + v_{\hat{t}+1} \cdot 1 \cdot \frac{\tilde{x}_{\hat{t}+1}}{\lceil z_{\hat{t}} \rceil - z_{\hat{t}}} \cdot (\lceil z_{\hat{t}} \rceil - z_{\hat{t}})$$

$$= U^{\hat{t}} + v_{\hat{t}+1} \cdot \tilde{x}_{\hat{t}+1}$$

$$= U^{\hat{t}+1}.$$

**Case 2 -** $\lceil z_{\hat{t}+1} \rceil \neq \lceil z_{\hat{t}} \rceil$**:** In this case, if item $\lceil z_{\hat{t}} \rceil$ was still available, it will be allocated with probability 1. If it is not available, item $\lceil z_{\hat{t}+1} \rceil$ will start to be allocated to this buyer. Therefore,

the expected utility in this case is:

$$\sum_{t=1}^{\hat{t}+1} \mathbb{E}[v_t \cdot x_t] = U^{\hat{t}} + v_{\hat{t}+1} \cdot \mathbb{E}[x_{\bar{t}+1}]$$

$$= U^{\hat{t}} + v_{\hat{t}+1} \cdot \left( 1 \cdot 1 \cdot (\lceil z_{\hat{t}} \rceil - z_{\hat{t}}) + 1 \cdot \frac{z_{\hat{t}+1} - \lceil z_{\hat{t}} \rceil}{1 - \lceil z_{\hat{t}} \rceil + z_{\hat{t}}} \cdot (1 - (\lceil z_{\hat{t}} \rceil - z_{\hat{t}})) \right)$$

$$= U^{\hat{t}} + v_{\hat{t}+1} \cdot (z_{\hat{t}+1} - z_{\hat{t}})$$

$$= U^{\hat{t}} + v_{\hat{t}+1} \cdot \tilde{x}_{\hat{t}+1}$$

$$= U^{\hat{t}+1}.$$

In both cases, we can see that the expected performance of Algorithm 1 remains the same as that of the fractional algorithm. Additionally, we know that Algorithm 5 achieves the optimal competitive ratio in the fractional setting . As a result, this algorithm can guarantee the same expected performance of the fractional optimal solution. We highlight that ROUNDING can preserve the performance of any fractional algorithm. We provide an example of such algorithm in Appendix B.3.

## B.3 An Example of Optimal Fractional Algorithm for OMcS with GFQ

Here we provide an example of an optimal fractional algorithm with GFQ constraint in the relaxed fractional setting in Algorithm 5. This algorithm leverages a singe threshold function to achieve the optimal competitive ratio. In particular, the algorithm reserves $M$ items to ensure the satisfaction of GFQ requirement where $M = \sum_{j \in [K]} m_i$ and then allocate the remaining $B - M$ items based on the threshold function $\phi(u)$. Let $C_j = B - \max_{i \in [j-1]} \{m_i\}$ and $D_j = \sum_{i=1}^{j-1} \theta_i \cdot [m_i - \max_{k \in [i-1]} \{m_k\}]^+$ where $[\cdot]^+ = \max\{\cdot, 0\}$. The design of the threshold function is based on the value of $M$. In particular, when $M \leq \frac{B}{\alpha_0^*}$, the threshold function $\phi$ is designed as

$$\phi(u) = \begin{cases} 1 & u \in [0, \Gamma^0], \\ \exp\left( \frac{\alpha_0^* \cdot (u+M) - B - \sum_{i=1}^{j-1} (C_i - C_{i+1}) \ln \theta_i}{C_j} \right) & u \in [\Gamma^{j-1}, \Gamma^j], \quad \forall j \in [K], \end{cases}$$

and is $\alpha_0^*$-competitive where $\alpha_0^* := 1 + \ln \theta_K - \sum_{j=1}^{K-1} \frac{(C_j - C_{j+1})}{B} \ln(\frac{\theta_K}{\theta_j})$ and $\Gamma^j = \frac{B}{\alpha_0^*} - M + \frac{C_j}{\alpha_0^*} \ln \theta_j + \frac{1}{\alpha_0^*} \sum_{i=1}^{j-1} (C_i - C_{i+1}) \ln \theta_i$. On the other hand, when $M \in \left( \frac{\theta_{j^*-1} \cdot C_{j^*} + D_{j^*}}{\alpha_{j^*}}, \frac{\theta_{j^*} \cdot C_{j^*} + D_{j^*}}{\alpha_{j^*}} \right]$ for some $j^* \in [K]$, the threshold function $\phi(u)$ is designed as

$$\phi(u) = \left\{ v^* \exp\left( \frac{\alpha_{j^*} \cdot u - \sum_{i=j^*}^{j-1} (C_i - C_{i+1}) \ln\left( \frac{\theta_i}{v^*} \right)}{C_j} \right) \quad u \in [\Gamma_{j^*}^{j-1}, \Gamma_{j^*}^j], \quad \forall j \in \{j^*, \cdots, K\}, \right.$$

and is $\alpha_{j^*}$-competitive where $\alpha_{j^*}$ is defined as

$$\alpha_{j^*} = \frac{D_{j^*}}{M} + \frac{C_{j^*}}{B - M} W\left( \frac{\theta_K (B - M)}{M} \exp\left( -\frac{X}{C_{j^*}} \right) \exp\left( -\frac{D_{j^*}(B-M)}{C_{j^*} \cdot M} \right) \right),$$

with $X = \sum_{i=j^*}^{K-1} (C_i - C_{i+1}) \cdot \ln\left( \frac{\theta_K}{\theta_i} \right)$, $v^* = (\alpha_{j^*} \cdot M - D_{j^*})/C_{j^*}$, and $\Gamma_{j^*}^j = \frac{C_j}{\alpha_{j^*}} \ln \frac{\theta_j}{v^*} + \frac{1}{\alpha_{j^*}} \sum_{i=j^*}^{j-1} (C_i - C_{i+1}) \ln \frac{\theta_i}{v^*}$. This algorithm builds on the fractional approach of [ZJST25]; for a comprehensive treatment of that method, we direct the reader to that work, since it lies beyond the primary scope of this paper.

**Algorithm 5:** Fractional OMcS with GFQ guarantee (FRAC-GFQ)

**Input:** $(m_j, \theta_j), \forall j \in [K]$.

**Initialization:** Initial global utilization, $u_0 = 0$; Initial utilization of class $j$, $u_0^j = 0, \forall j \in [K]$.

**1 while** *agent $t$ arrives* **do**

**2**      Obtain the valuation and class information of agent $t$: $v_t$ and $\mathcal{J}_t$;

**3**      **if** $u_{t-1}^{j_t} < m_{j_t}$ **then**             ▷ Satisfying GFQ constraint.

**4**          $y_t = \min\{r_t, m_{j_t} - u_{t-1}^{j_t}\}$.

**5**          Update $u_t^{j_t} = u_{t-1}^{j_t} + y_t$.

**6**      **if** $v_t \geq \phi(u_{t-1})$ **then**           ▷ Allocating the remaining resource.

**7**          $x_t = \min\left\{\arg\max_{a \in [0, r_t - y_t]} \left\{ av_t - \int_{u_{t-1}}^{u_{t-1}+a} \phi(\eta)d\eta \right\}, B - M - u_{t-1}\right\}$.

**8**      Update the cumulative allocation: $u_t = u_{t-1} + x_t$.

**9**      Update the allocation amount of agent $t$: $x_t = x_t + y_t$.

## C    Section 4 Proofs

### C.1    Proof of Theorem 4.1

Let $\mathbf{x}$ denote R-SETASIDE-PF's final allocation. Here we consider the following relaxed LP and its dual which does not have any allocation limit at each time step:

(Primal)

$$\max_{\mathbf{w} \in \mathbb{R}_+^T} \frac{1}{K} \sum_{i=1}^{K} \frac{U_i(\mathbf{w})}{U_i(\mathbf{x})}$$

$$\text{s.t.} \quad \sum_{t=1}^{T} w_t \leq B$$

(Dual)

$$\min_{q \in \mathbb{R}_+} \; B\,q$$

$$\text{s.t.} \quad q \geq \frac{1}{K} \sum_{i \in \mathcal{J}_t} \frac{v_t}{U_i(\mathbf{x})}, \; \forall t$$

Now, show that $\beta(\mathfrak{b})/B$ is a feasible solution for the above dual LP. We can see that at time $t$, $U_i(\mathbf{x}) \geq \sum_{j \in \mathcal{J}_t} \Phi_{i,j}(v_t)$, where $\Phi_{i,j}(v) = \Upsilon_{i,j}(1) + \int_{\Upsilon_{i,j}(1)}^{\Upsilon_{i,j}(v)} \phi_{i,j}(u)du$, where $\Upsilon_{i,j}(v)$ for each $v \in [1, \min \theta_i, \theta_j]$ as follows:

$$\Upsilon_{i,j}(v) = \arg\max_{a \geq 0} \left( a \cdot v - \int_0^a \phi_{i,j}(u)\, du \right).$$

As a result, it is easy to see that:

$$U_i(\mathbf{x}) \geq \Phi_{i,i}(v_t) + \sum_{j \in \mathcal{J}_t, j \neq i} \Phi_{i,j}(v_t) + \Phi_G(v_t)$$

$$\geq \Phi_{i,i}(v_t) + \sum_{j \in \mathcal{J}_t, j \neq i} \Phi_{i,j}(v_t)$$

$$\geq \left[ \frac{B}{K \cdot \beta(\mathfrak{b})} + \frac{B}{K \cdot \beta(\mathfrak{b})} \cdot (v_t - 1) \right] \cdot (1 + |\mathcal{J}_t| - 1)$$

$$\geq \frac{B \cdot v_t \cdot \mathcal{J}_t}{K \cdot \beta(\mathfrak{b})}.$$

Therefore, we can easily see that:

$$\frac{\beta(\mathfrak{b})}{B} \geq \frac{1}{K} \sum_{i \in \mathcal{J}_t} \frac{v_t}{\frac{B \cdot v_t \cdot \mathcal{J}_t}{K \cdot \beta(\mathfrak{b})}} \geq \frac{1}{K} \sum_{i \in \mathcal{J}_t} \frac{v_t}{U_i(\mathbf{x})}.$$

Then by weak duality we will have:

$$\max_{\mathbf{w} \in \mathbb{R}_+^T} \frac{1}{K} \sum_{i=1}^{K} \frac{U_i(\mathbf{w})}{U_i(\mathbf{x})} \leq B \cdot \frac{\beta(\mathfrak{b})}{B} = \frac{1}{1 - \mathfrak{b}} \cdot \sum_{j \in [K]} \left(1 - \frac{j-1}{K}\right) \cdot \alpha_j$$

Additionally, we note that in this case, $\mathsf{OPT}$, the revenue of optimal offline algorithm can be lower bounded as follows

$$\mathsf{OPT} \leq B \cdot v_t.$$

On the other hand, $\mathsf{ALG}$, the objective of the Algorithm 3, is lower bounded by

$$\mathsf{ALG} \geq \sum_{i,j \in \mathcal{J}_t} \Phi_{i,j}(v_t) + \Phi_G(v_t)$$

$$\geq \left[ \frac{B}{K \cdot \beta(\mathfrak{b})} + \frac{B}{K \cdot \beta(\mathfrak{b})} \cdot (v_t - 1) \right] \cdot \left( |\mathcal{J}_t| + \binom{|\mathcal{J}_t|}{2} \right) + \frac{B \cdot \beta(\mathfrak{b})}{\alpha_K} \cdot v_t$$

$$\geq \left[ \frac{B}{K \cdot \beta(\mathfrak{b})} + \frac{B}{K \cdot \beta(\mathfrak{b})} \cdot (v_t - 1) \right] + \frac{B \cdot \beta(\mathfrak{b})}{\alpha_K} \cdot v_t$$

$$\geq B \cdot v_t \cdot \left[ \frac{1}{K \cdot \beta(\mathfrak{b})} + \frac{\beta(\mathfrak{b})}{\alpha_K} \right]$$

$$\geq \mathsf{OPT} \cdot \left[ \frac{1}{K \cdot \beta(\mathfrak{b})} + \frac{\beta(\mathfrak{b})}{\alpha_K} \right]$$

$$= \mathsf{OPT} \cdot \frac{1}{\alpha(\mathfrak{b})}.$$

This completes the proof of Theorem 4.3 in the relaxed version. We point out that, based on the results of [ZJST25, SZL$^+$20, LCS$^+$23], the performance guarantee is also preserved in the constrained version with $x_t \leq 1$ for all $t \in [T]$.

### C.2 Proof of Corollary 4.2

When $|\mathcal{J}_t| = 1$ for all $t \in [T]$, each arriving agent is said to be single-labeled. In this case, the reservation budgets associated with pairwise threshold functions become redundant, and we can safely set $b_{i,j} = 0$ for all $i \neq j$, $i, j \in [K]$. Consequently, the resulting threshold function design reduces to the one proposed by [ZJST25], which is known to achieve the Pareto-optimal trade-off between fairness and efficiency. We refer to this paper for the full discussion about this trade-off.

### C.3 Proof of Theorem 4.3

To establish that Algorithm 3 maintains its expected performance after the rounding step at every time step, we apply mathematical induction. Define the cumulative utility accrued by class $i$ up to time $t$ under the fractional allocation as $U_i^t = \sum_{t'=0}^{t} v_{t'} \cdot \tilde{x}_{t'} \cdot \mathbf{1}_{\{i \in \mathcal{J}_{t'}\}}$. We start with the base case. Let $\bar{t}$ be the first time index such that $\tilde{x}_{\bar{t}} \in (0, 1]$, indicating a non-integral allocation by the fractional algorithm at that step. At this point, the algorithm assigns $\kappa_{\bar{t}} = \lceil z_{\bar{t}} \rceil$, triggering a randomized rounding. As a result, the expected utility after rounding step is

$$\sum_{t=1}^{\bar{t}} \mathbb{E}[v_t \cdot x_t \cdot \mathbf{1}_{\{i \in \mathcal{J}_t\}}] = v_{\bar{t}} \cdot \mathbb{E}[x_{\bar{t}} \cdot \mathbf{1}_{\{i \in \mathcal{J}_t\}}] = v_{\bar{t}} \cdot (1 \cdot \mathbf{1}_{\{i \in \mathcal{J}_t\}} \cdot (z_{\bar{t}} - \lceil z_{\bar{t}-1} \rceil))$$

$$= v_{\bar{t}} \cdot (1 \cdot \mathbf{1}_{\{i \in \mathcal{J}_t\}} \cdot (z_{\bar{t}} - z_{\bar{t}-1})) = v_{\bar{t}} \cdot \tilde{x}_{\bar{t}} \cdot \mathbf{1}_{\{i \in \mathcal{J}_t\}} = U_i^{\bar{t}}.$$

To complete the proof, it remains to verify the induction step. Suppose that up to time $\hat{t}$, the expected performance of Algorithm 3 aligns with the fractional utility, i.e., $\sum_{t=1}^{\hat{t}} \mathbb{E}[v_t \cdot x_t \cdot \mathbf{1}_{\{i \in \mathcal{J}_t\}}] = U_i^{\hat{t}}$. We aim to show that this equality holds at time $\hat{t} + 1$ as well ($\sum_{t=1}^{\hat{t}+1} \mathbb{E}[v_t \cdot x_t \cdot \mathbf{1}_{\{i \in \mathcal{J}_t\}}] = U_i^{\hat{t}+1}$). To do so, we analyze the availability of item $\lceil z_{\hat{t}} \rceil$ at time $\hat{t}$. According to the rounding scheme, the probability that this item is still available is given by $\lceil z_{\hat{t}} \rceil - z_{\hat{t}}$. This expression is derived inductively by examining the cumulative allocation and rounding behavior. We now proceed by explicitly computing this probability as follows:

$$P(\text{item } \lceil z_{\hat{t}} \rceil \text{ available}) = P(\text{item } \lceil z_{\hat{t}} \rceil \text{ available} | \text{item } \lceil z_{\hat{t}-1} \rceil \text{ available}) \cdot P(\text{item } \lceil z_{\hat{t}-1} \rceil \text{ available}) +$$
$$P(\text{item } \lceil z_{\hat{t}} \rceil \text{ available} | \text{item } \lceil z_{\hat{t}-1} \rceil \text{ not available}) \cdot P(\text{item } \lceil z_{\hat{t}-1} \rceil \text{ not available}).$$

Now in the case $\lceil z_{\hat{t}} \rceil = \lceil z_{\hat{t}-1} \rceil$, the probability of item $\lceil z_{\hat{t}-1} \rceil$ being available is 0. As a result, we have

$$P(\text{item } \lceil z_{\hat{t}} \rceil \text{ available}) = \left(1 - \frac{\tilde{x}_{\hat{t}-1}}{\lceil z_{\hat{t}-1} \rceil - z_{\hat{t}-1}}\right) \cdot (\lceil z_{\hat{t}-1} \rceil - z_{\hat{t}-1}) + 0$$

$$= \lceil z_{\hat{t}-1} \rceil - z_{\hat{t}-1} - \tilde{x}_{\hat{t}-1} = \lceil z_{\hat{t}} \rceil - z_{\hat{t}}.$$

When $\lceil z_{\hat{t}} \rceil \neq \lceil z_{\hat{t}-1} \rceil$ we have

$$P(\text{item } \lceil z_{\hat{t}} \rceil \text{ available}) = 1 \cdot (\lceil z_{\hat{t}-1} \rceil - z_{\hat{t}-1}) + \left(1 - \frac{z_{\hat{t}} - \lceil z_{\hat{t}-1} \rceil}{1 - \lceil z_{\hat{t}-1} \rceil + z_{\hat{t}-1}}\right) \cdot (1 - \lceil z_{\hat{t}-1} \rceil + z_{\hat{t}-1})$$

$$= \lceil z_{\hat{t}-1} \rceil - z_{\hat{t}-1} + 1 + z_{\hat{t}-1} - z_{\hat{t}} = \lceil z_{\hat{t}-1} \rceil + 1 - z_{\hat{t}} = \lceil z_{\hat{t}} \rceil - z_{\hat{t}}.$$

This concludes that at each time $\hat{t}$, item $\lceil z_{\hat{t}} \rceil$ is available with probability $\lceil z_{\hat{t}} \rceil - z_{\hat{t}}$. Additionally, since the utilities are linear we have $\sum_{t=1}^{\hat{t}+1} \mathbb{E}[v_t \cdot x_t \cdot \mathbf{1}_{\{i \in \mathcal{J}_t\}}] = U^{\hat{t}} + v_{\hat{t}+1} \cdot \mathbb{E}[x_{\hat{t}+1} \cdot \mathbf{1}_{\{i \in \mathcal{J}_t\}}]$.

Based on the algorithm, there are two possible cases regarding the allocated item at time $\hat{t}+1$:

**Case 1 -** $\lceil z_{\hat{t}+1} \rceil = \lceil z_{\hat{t}} \rceil$**:** In this case the only possibility is to allocate item $\lceil z_{\hat{t}} \rceil$. This item is available with probability $\lceil z_{\hat{t}} \rceil - z_{\hat{t}}$. Therefore, the expected utility in this case is:

$$\sum_{t=1}^{\hat{t}+1} \mathbb{E}[v_t \cdot x_t \cdot \mathbf{1}_{\{i \in \mathcal{J}_t\}}] = U_i^{\hat{t}} + v_{\hat{t}+1} \cdot \mathbb{E}[x_{\bar{t}+1} \cdot \mathbf{1}_{\{i \in \mathcal{J}_t\}}]$$

$$= U_i^{\hat{t}} + v_{\hat{t}+1} \cdot 1 \cdot \frac{\tilde{x}_{\hat{t}+1} \cdot \mathbf{1}_{\{i \in \mathcal{J}_t\}}}{\lceil z_{\hat{t}} \rceil - z_{\hat{t}}} \cdot (\lceil z_{\hat{t}} \rceil - z_{\hat{t}})$$

$$= U_i^{\hat{t}} + v_{\hat{t}+1} \cdot \tilde{x}_{\hat{t}+1} \cdot \mathbf{1}_{\{i \in \mathcal{J}_t\}}$$

$$= U_i^{\hat{t}+1}.$$

**Case 2 -** $\lceil z_{\hat{t}+1} \rceil \neq \lceil z_{\hat{t}} \rceil$**:** In this case, if item $\lceil z_{\hat{t}} \rceil$ was still available, it will be allocated with probability 1. If it is not available, item $\lceil z_{\hat{t}+1} \rceil$ will start to be allocated to this buyer. Therefore, the expected utility in this case is:

$$\sum_{t=1}^{\hat{t}+1} \mathbb{E}[v_t \cdot x_t \cdot \mathbf{1}_{\{i \in \mathcal{J}_t\}}] = U_i^{\hat{t}} + v_{\hat{t}+1} \cdot \mathbb{E}[x_{\bar{t}+1} \cdot \mathbf{1}_{\{i \in \mathcal{J}_t\}}]$$

$$= U_i^{\hat{t}} + v_{\hat{t}+1} \cdot \left(1 \cdot 1 \cdot (\lceil z_{\hat{t}} \rceil - z_{\hat{t}}) + 1 \cdot \frac{z_{\hat{t}+1} - \lceil z_{\hat{t}} \rceil}{1 - \lceil z_{\hat{t}} \rceil + z_{\hat{t}}} \cdot (1 - (\lceil z_{\hat{t}} \rceil - z_{\hat{t}}))\right)$$

$$= U_i^{\hat{t}} + v_{\hat{t}+1} \cdot (z_{\hat{t}+1} - z_{\hat{t}})$$

$$= U_i^{\hat{t}} + v_{\hat{t}+1} \cdot \tilde{x}_{\hat{t}+1} \cdot \mathbf{1}_{\{i \in \mathcal{J}_t\}}$$

$$= U_i^{\hat{t}+1}.$$

In both cases, we can see that the expected performance after rounding step is similar to the relaxation step for each class $i \in [K]$. Additionally, with the same analysis we can proof that the expected total utility is also equal to the fractional part of the algorithm. As a result, it proofs that the fairness-efficiency trade-off is also same as the fractional case which is presented in Theorem 4.1.

# D  Section 5 Proofs

## D.1  Proof of Theorem 5.1

For a fixed instance $I$, let $U_i(\mathbf{x})$, $U_i(\hat{\mathbf{x}})$ and $U_i(\bar{\mathbf{x}})$ be the expected utility of class $i$ in LiLA, ADV and ALG, respectively. Then it can be seen that $U_i(\mathbf{x}) = \rho \cdot U_i(\hat{\mathbf{x}}) + (1 - \rho) \cdot U_i(\bar{\mathbf{x}})$. Let us define the function $f(p) = \frac{1}{p}$. Now let us consider $p_j = \rho + (1 - \rho)\frac{U_j(\bar{\mathbf{x}})}{U_j(\hat{\mathbf{x}})}$ and based on the Jensen's inequality we have

$$\frac{1}{K} \sum_{j \in [K]} \frac{1}{\rho + (1 - \rho)\frac{U_j(\bar{\mathbf{x}})}{U_j(\hat{\mathbf{x}})}} \leq \frac{1}{\rho + (1 - \rho)\frac{1}{K}\sum_{j \in [K]}\frac{U_j(\bar{\mathbf{x}})}{U_j(\hat{\mathbf{x}})}}.$$

Additionally, we can see

$$\frac{K}{\sum_{j\in[K]}\frac{U_j(\bar{\mathbf{x}})}{U_j(\hat{\mathbf{x}})}} \leq \frac{1}{K}\sum_{j\in[K]}\frac{U_j(\hat{\mathbf{x}})}{U_j(\bar{\mathbf{x}})} \leq \beta,$$

where the first inequality is based on the harmonic mean-arithmetic mean (HM-AM) inequality and the second one is base on the definition of $\beta$-PF. As a result we can see that

$$\frac{1}{K}\sum_{j\in[K]}\frac{1}{\rho+(1-\rho)\frac{U_j(\bar{\mathbf{x}})}{U_j(\hat{\mathbf{x}})}} \leq \frac{1}{\rho+(1-\rho)\frac{1}{K}\sum_{j\in[K]}\frac{U_j(\bar{\mathbf{x}})}{U_j(\hat{\mathbf{x}})}} \leq \frac{1}{\rho+(1-\rho)\frac{1}{\beta}},$$

which implies

$$\frac{1}{K}\sum_{j\in[K]}\frac{U_j(\hat{\mathbf{x}})}{U_j(\mathbf{x})} = \frac{1}{K}\sum_{j\in[K]}\frac{U_j(\hat{\mathbf{x}})}{\rho\cdot U_j(\hat{\mathbf{x}})+(1-\rho)U_j(\bar{\mathbf{x}})} \leq \frac{\beta}{\rho\cdot\beta+(1-\rho)} = 1+\epsilon.$$

Now let us consider $p_j = \rho\cdot\frac{U_j(\hat{\mathbf{x}})}{U_j(\mathbf{w})} + (1-\rho)\cdot\frac{U_j(\bar{\mathbf{x}})}{U_j(\mathbf{w})}$, where $\mathbf{w}$ is any feasible allocation. Using the Jensen's inequality we obtain

$$\frac{1}{K}\sum_{j\in[K]}\frac{1}{\rho\cdot\frac{U_j(\hat{\mathbf{x}})}{U_j(\mathbf{w})}+(1-\rho)\cdot\frac{U_j(\bar{\mathbf{x}})}{U_j(\mathbf{w})}} \leq \frac{1}{\rho\cdot\frac{1}{K}\sum_{j\in[K]}\frac{U_j(\hat{\mathbf{x}})}{U_j(\mathbf{w})}+(1-\rho)\cdot\frac{1}{K}\sum_{j\in[K]}\frac{U_j(\bar{\mathbf{x}})}{U_j(\mathbf{w})}}.$$

Since ADV does not have any constraint, therefore it is only possible to obtain $\frac{1}{K}\sum_{j\in[K]}\frac{U_j(\hat{\mathbf{x}})}{U_j(\mathbf{w})} \geq 0$. Additionally, based on the definition of $\beta$-PF and HM-AM inequality, we have $\frac{1}{K}\sum_{j\in[K]}\frac{U_j(\bar{\mathbf{x}})}{U_j(\mathbf{w})} \geq \frac{1}{\beta}$. As a result, we have that

$$\frac{1}{K}\sum_{j\in[K]}\frac{1}{\rho\cdot\frac{U_j(\hat{\mathbf{x}})}{U_j(\mathbf{w})}+(1-\rho)\cdot\frac{U_j(\bar{\mathbf{x}})}{U_j(\mathbf{w})}} \leq \frac{1}{(1-\rho)\cdot\frac{1}{\beta}},$$

which implies

$$\frac{1}{K}\sum_{j\in[K]}\frac{U_j(\mathbf{w})}{U_j(\mathbf{x})} = \frac{1}{K}\sum_{j\in[K]}\frac{U_j(\mathbf{x})}{\rho\cdot U_j(\hat{\mathbf{x}})+(1-\rho)U_j(\bar{\mathbf{x}})} \leq \frac{\beta}{1-\rho} = \frac{(1+\epsilon)(\beta-1)}{\epsilon}.$$

This concludes the consistency and robustness proportional fairness guarantee presented in Theorem 5.1.

## D.2 Proof of Pareto-optimality of Consistency-Robustness in Single-labeled Setting

To demonstrate the Pareto-optimality result when $|\mathcal{J}_t| = 1$, we first establish that the relaxed LiLA —a linear combination of the robust decision and the advice—is Pareto optimal. We first construct a hard instance and then show that for any $\gamma$-robust learning augmented algorithm, their consistency $\eta$ is lower bounded under the special instances.

**Definition D.1** ($\beta$-PF Fairness Guarantee Hard Instance: $I^{\mathsf{PF}}$). Instance $I^{\mathsf{PF}}$ is defined as a scenario characterized by a at most $K$ continuous, non-decreasing sequence of valuation arrivals segments. In this scenario, first there are a sequence of arrivals from class 1, followed by the second sequence of arrivals all from class 2 and and this continues until the arrivals of class $K$. For some value of $\delta$ such that $\delta \to 0$, instance $I^{\mathsf{PF}}$ can be shown as follows:

$$I^{\mathsf{PF}} = \left\{ \underbrace{(1,1),(1+\delta,1),\ldots,(\theta_1,1)}_{\text{First batch of arrivals}}, \underbrace{(1,2),(1+\delta,2),\ldots,(\theta_2,2)}_{\text{Second batch of arrivals}}, \ldots, \right.$$

$$\left. \underbrace{(1,K),(1+\delta,K),\ldots,(\theta_K,K)}_{K\text{-th batch of arrivals}} \right\},$$

where in above $(v,j), \forall j\in[k]$, corresponds to the $B$ copies of a buyer with valuation equal to $v$ from class $j$.

Let $g_j(p) : [1, \theta_j] \to [0, b_j]$ denote a non-decreasing utilization function of any learning augmented algorithm for OMcS problem. A key observation is that for a small $\delta$, executing the instance $I^{PF}$ up to the $\boldsymbol{v}(c_j)$ is equivalent to first executing $I^{PF}$ to the $\boldsymbol{v} - \boldsymbol{\delta}(c_j)$ (excluding the last step) and then processing $\boldsymbol{v}(c_j)$ for some class $j \in [K]$. Additionally, since due to the budget constraint of each class, we can see that $g_j(\theta_j) \leq b_j$.

Let us first consider the case where the stoping poin is at some $v$ from class 1. Then, for any $\gamma$-robust proportionally fair online algorithm the PF condition simplifies to:

$$\frac{1}{K} \cdot \frac{U_1(\mathbf{w})}{U_1(\mathbf{x})} \leq \frac{1}{K} \cdot \frac{B \cdot v}{g_1(1) + \int_1^v u \, dg_1(u)} \leq \gamma.$$

By integral by parts and the Gronwall's inequality, a necessary condition for the above robustness constraint to hold is

$$g_1(v) \geq \frac{B}{K \cdot \gamma} \cdot (1 + \ln \theta_1).$$

In addition, to ensure $\eta$-consistency when the prediction is accurate and $v = \theta_1$, we must ensure $\frac{1}{K} \cdot \frac{U_1(\mathbf{w})}{U_1(\mathbf{x})} \leq \eta$. Combining this constraint with $g_i(\theta_1) \leq b_1$ gives

$$\frac{1}{K} \cdot \frac{U_1(\mathbf{w})}{U_1(\mathbf{x})} \leq \frac{1}{K} \cdot \frac{B \cdot \theta_1}{g_1(1) + \int_1^{\theta_1} u \, dg_1(u) + (b_1 - g_1(\theta_1)) \cdot \theta_1} \leq \eta,$$

where $(b_1 - g_1(\theta_1))$ is the portion of $b_1$ that is remaining to ensuring the consistency. The above implies that

$$b_1 \geq \frac{B}{K \cdot \eta} + \frac{B \cdot \ln \theta_1}{K \cdot \gamma}.$$

By executing $I^{PF}$ for other classes, we can see that $b_{i,j} \geq \frac{B}{K \cdot \eta} + \frac{B \cdot \ln \theta_j}{K \cdot \gamma}$. Therefore by summing up for all class $i, j \in [K]$ and since $B \leq \sum_{i,j \in [K]} b_{i,j}$, we can see that

$$B \geq \sum_{i,j \in [K]} b_{i,j} \geq \sum_{i,j \in [K]} \left( \frac{B}{K \cdot \eta} + \frac{B \cdot \ln(\min\{\theta_i, \theta_j\})}{K \cdot \gamma} \right) = B \cdot \left( \frac{K+1}{2 \cdot \eta} + (\beta - 1) \cdot \frac{1}{\gamma} \right),$$

where $\beta = \frac{\sum_{j \in [K]} (K - j + 1) \cdot \alpha_j}{K}$ is the proportional fairness of Algorithm 3. Now by setting $\eta = 1 + \epsilon$, we obtain that

$$\gamma \geq \frac{(\beta - 1)(1 + \epsilon)}{\epsilon}.$$

This result states that for any $(1 + \epsilon)$-consistent proportionally fair algorithm, the robustness is at least $\frac{(\beta-1)(1+\epsilon)}{\epsilon}$, which concludes the proof of the Pareto optimality. Additionally, using the same approach as Algorithm 3 to round the decisions, we can have the Pareto optimality result for the integral LiLA as well.

### D.3 Proof of Corollary 5.2

Based on the proof of Theorem 5.1 we know that any $\eta$-consistent and $\gamma$-robust proportionally fair algorithm must reserve $b_j = \frac{B}{K \cdot \eta} + \frac{B \cdot \ln \theta_j}{K \cdot \gamma}$ for each class. As a result, under some instance $I$ that all the valuations are from some class $j \in [K]$ with the maximum value $v$, we can see that $\mathsf{OPT}(I) \leq B \cdot v$. On the other hand LiLA can obtain $\mathsf{LiLA}(I) \geq g_j(1) + \int_1^v u \, dg_j(u)$ for any $v \in [1, \theta_j)$. To ensure the $\gamma_\alpha$-robustness in terms of competitiveness, it is essential to satisfy

$$g_j(1) + \int_1^v u \, dg_1(u) \geq \frac{1}{\gamma_\alpha} \cdot B \cdot v.$$

Now based on the design of $g_j(\cdot)$ that can achieve $\gamma$-robustness proportional fairness, we have

$$\frac{B}{K \cdot \gamma} \geq \frac{B}{\gamma_\alpha} \cdot v,$$

which implies $\gamma_\alpha \geq K \cdot \gamma = K \cdot \left( \frac{(1+\epsilon)(\beta-1)}{\epsilon} \right)$. Now when the prediction is accurate and $v = \theta_j$, we have

$$g_j(1) + \int_1^{\theta_j} u \, dg_j(u) + (b_j - g_j(\theta_j))\theta_j \geq \frac{1}{\eta_\alpha} \cdot B \cdot \theta_j,$$

which based on the reservation $b_j = \frac{B}{K \cdot \eta} + \frac{B \cdot \ln \theta_j}{K \cdot \gamma}$ implies that $\eta_\alpha \geq K \cdot \eta = K \cdot (1 + \epsilon)$. This concludes that any $\eta$-consistent and $\gamma$-robust proportionally fair algorithm is at least $\eta_\alpha$-consistent and $\gamma_\alpha$-robust competitive with $\eta_\alpha = K \cdot \eta$ and $\gamma_\alpha = K \cdot \gamma$.

### D.4 Learning-Augmented Algorithm for OMcS with GFQ.

Here we consider a learning-augmented algorithm that utilizes untrusted machine learning-generated advice to improve the performance of robust algorithms for OMcS with GFQ guarantee.

**Definition D.2** (Advice model of OMcS with GFQ). For the OMcS with GFQ fairness guarantee, we denote $\mathsf{ADV} := \{\hat{x}_t \in \{0, 1\} | \sum_{t \in [T]} \hat{x}_t \cdot \mathbf{1}_{\{j \in \mathcal{J}_t\}} \geq m_j, \forall j \in [K]\}$ as the untrusted black-box decision advice.

The first observation we can make is that it is assumed ADV always satisfies the GFQ constraint. Additionally if the advice is completely correct, it basically recovers the optimal offline decisions. Here we again use LiLA which combines the robust decision (i.e., $\bar{x}_t$) that is resulted from Algorithm 1 at each time step and the predicted optimal solution (i.e., $\hat{x}_t$) generated by black-box advice with a combination probability $\rho$ that shows the reliance on the advice decision. As a result, the decision of the learning augmented algorithm at each time in expectation is $x_t = \rho \hat{x}_t + (1 - \rho)\bar{x}_t$.

For an $\epsilon \in [0, \alpha - 1]$, where $\alpha$ is the competitive ratio of Algorithm 1, we sets the combination probability as $\rho := (\frac{\alpha}{1+\epsilon} - 1) \cdot \frac{1}{\alpha-1}$ which is in $[0, 1]$. Here we can observe that as the prediction error $\epsilon$ approaches to 0, $\rho$ approaches to 1, which means that the algorithm fully trusts the prediction. Theorem D.1 below shows our main results of the learning augmented algorithm of OMcS with GFQ constraints.

**Theorem D.1** (Learning-augmented algorithm for OMcS with GFQ). *For any $\epsilon \in [0, \alpha - 1]$, LiLA for OMcS with GFQ constraint is $(1 + \epsilon)$-consistent and $\left( \frac{(1+\epsilon)(\alpha-1)}{\epsilon+(\alpha-1-\epsilon) \cdot \frac{M}{C_K \theta_K + D_K}} \right)$-robust.*

*Proof.* We begin by pointing out that the online solution given by LiLA is always feasible in expectation:

$$\sum_{t \in [T]} x_t \cdot \mathbf{1}_{\{j \in \mathcal{J}_t\}} = \rho \cdot \sum_{t \in [T]} \hat{x}_t \cdot \mathbf{1}_{\{j \in \mathcal{J}_t\}} + (1 - \rho) \cdot \sum_{t \in [T]} \bar{x}_t \cdot \mathbf{1}_{\{j \in \mathcal{J}_t\}}$$
$$\geq \rho \cdot m_j + (1 - \rho) \cdot m_j \geq m_j,$$

which is true since ADV always produces a feasible advice.

For any instance $I$, we can see that

$$\mathsf{LiLA}(I) = \sum_{t \in [K]} v_t x_t = \sum_{t \in [K]} v_t \cdot (\rho \hat{x}_t + (1 - \rho)\bar{x}_t)$$
$$= \rho \sum_{t \in [K]} v_t \hat{x}_t + (1 - \rho) \sum_{t \in [K]} v_t \bar{x}_t = \rho \cdot \mathsf{ADV}(I) + (1 - \rho)\mathsf{ALG}(I).$$

Based on the definition of competitive ratio we can see that

$$\frac{\mathsf{ADV}(I)}{\mathsf{ALG}(I)} \leq \frac{\mathsf{OPT}(I)}{\mathsf{ALG}(I)} \leq \alpha^*.$$

As a result, $\frac{1}{\alpha^*} \cdot \mathsf{ADV}(I) \leq \mathsf{ALG}(I)$. Therefore

$$\mathsf{LiLA}(I) \geq (\rho + \frac{1}{\alpha^*}(1 - \rho))\mathsf{ADV}(I) = \frac{\rho \alpha^* + (1 - \rho)}{\alpha^*} \cdot \mathsf{ADV}(I),$$

which implies

$$\frac{\mathsf{ADV}(I)}{\mathsf{LiLA}(I)} \leq \frac{\alpha^*}{\rho \alpha^* + (1 - \rho)} = (1 + \epsilon).$$

On the other hand, we have $\mathsf{ADV}(I) \geq \frac{M}{C_K \theta_K + D_K} \mathsf{OPT}(I)$ and $\mathsf{ALG}(I) \geq \frac{1}{\alpha^*} \mathsf{OPT}(I)$. Therefore,

$$\mathsf{LiLA}(I) \geq \rho \cdot \frac{M}{C_K \theta_K + D_K} \cdot \mathsf{OPT}(I) + (1-\rho) \cdot \frac{1}{\alpha^*} \cdot \mathsf{OPT}(I)$$
$$= \left( \rho \cdot \frac{M}{C_K \theta_K + D_K} + (1-\rho) \cdot \frac{1}{\alpha^*} \right) \cdot \mathsf{OPT}(I).$$

As a result

$$\frac{\mathsf{OPT}(I)}{\mathsf{LiLA}(I)} \leq \frac{(1+\epsilon)(\alpha^* - 1)}{\epsilon + (\alpha^* - 1 - \epsilon) \cdot \frac{M}{C_K \theta_K + D_K}}.$$

By combining these two, the consistency and robustness of Theorem D.1 follows. $\qquad\square$

