# OpenReview forum: "Online Multi-Class Selection with Group Fairness Guarantee"
_NeurIPS.cc/2025/Conference — NeurIPS 2025 poster_

### Official Review · Reviewer_fyob · 2025-06-30

**Clarity:** 3
**Significance:** 3
**Originality:** 2
**Rating:** 4
**Confidence:** 3

**Summary:**

The authors study the problem of online muti-class selection, where agents are arriving one at a time and based on each agent’s value, the algorithm must determine whether to allocate a resource to each agent. The goal is to allocate these resources ‘fairly’ to the agents by group, where each agent can be a member of multiple groups. The authors study two natural notions of fairness, Group Fairness by Quantity (guaranteeing a specific quantity to every group) and \beta-proportional fairness (guaranteeing a form of approximate proportional fairness for each group). The authors present a series of deterministic and randomized algorithms that solve the multi-class selection problem with these two notions of fairness, proving that for GFQ they achieve the optimal competitive ratio. The authors also provide some results of augmenting the algorithms with machine learning predictions.

**Questions:**

1. Can you describe in detail how the results and techniques in this paper differ from the most closely related works (namely [JZST24] and [ZJST25])?

2. Are there any specific settings or applications where the authors believe this algorithm could be useful or in which the new setting additions (multigroup, integral) would be especially useful?

**Ethical Concerns:**

["NO or VERY MINOR ethics concerns only"]

**Final Justification:**

I maintain that the problem and results are relatively interesting to the theory community.

**Limitations:**

Yes

**Quality:**

3

**Strengths And Weaknesses:**

Strengths
- The extension of group fairness to the multi-class setting is a natural question, as is the question of finding integral solutions rather than fractional solutions
- The algorithm techniques of rounding seem to be novel compared to the literature on this topic and the concepts are interesting and may generalize beyond the scope of this problem.
- The fairness notions studied in the paper are natural for the problem and are well-motivated by the authors
- While the writing is a bit dense at times, the paper is overall well-written

Weaknesses
- The work builds off of two related works [JZST24] and [ZJST25] that study the same problem with slightly different variations (studying single-labeled groups among other differences). While the multi-group setting is natural, it is not obvious to me that the advancements in this paper are interesting enough to the general machine learning for this conference. More specifically, I feel that the problem is somewhat specific and the improvements in this paper are somewhat incremental on previous works.
- This work is primarily a theoretical contribution, however it could be interesting to motivate with some applications of this problem and the authors' algorithm.

Minor presentation comments
- K seems to be reused on line 142 to be different than the K on line 118
- What is the definition of  \beta-NSW on line 168?
- It would be helpful if FRAC-GFQ is presented in the main body
-Would be helpful to define/explain what \kappa_i is for algorithm 1
- The description of algorithm 1 doesn’t quite match the algorithm, so would be helpful to clarify the description
- Could be helpful to have pseudocode given the significant amount of algorithm-specific notation
- When presenting LILA, please link to the appendix where the algorithm is formally presented

---

> ### Author Rebuttal · Authors · 2025-07-31
>
> Thank you for your insightful review and constructive feedback. Below, we address your comments one by one:
>
> *“Comparison with Closely Related Works ([JZST24] and [ZJST25]):”*
>
> Our work differs significantly from [JZST24] and [ZJST25] in several key aspects. Firstly, we address the allocation problem in an integral setting, whereas the mentioned works focus on fractional settings. This distinction, while subtle, is crucial, especially when dealing with limited initial inventory. Many real-world applications, such as vaccine and housing allocation, inherently require integral decisions since resources like houses cannot be fractionally divided among recipients. To bridge this gap, we introduced a novel rounding scheme that converts fractional solutions into integral ones without any loss in performance. Moreover, this rounding scheme is agnostic to how the fractional solutions are obtained. Thus, the lossless rounding technique is of independent interest. We also anticipate that this approach can be applied or extended to a broader range of application areas.
>
> Additionally, [JZST24] and [ZJST25] explore scenarios with single-labeled arrivals; however, we generalize the setting to multi-labeled arrivals. This generalization introduces additional layers of uncertainty, significantly complicating the analysis. Specifically, the multi-labeled setting involves uncertainties in the number of labels, valuation, and class information of arrivals, thus making the problem substantially more challenging.
>
>
>
> *“Specific Applications and Settings:”*
>
> Indeed, our algorithm is highly applicable to numerous real-world scenarios, especially for scenarios involving public goods allocation. For example, consider a shared caching buffer used by websites categorized by language (English, Spanish, etc.) and content type (news, gaming, entertainment, etc.). Each incoming request carries multiple labels reflecting these categories. In this scenario, our algorithm effectively ensures fair caching management among distinct user groups. Another interesting application of our model and algorithms is in the public housing allocation domain, where applicants may belong to one or multiple groups, depending on how groups are decided in the first place (e.g., based on race, gender, age, income, etc). In this case, the multi-group and integral allocation become essential. We will revise the introduction of the manuscript to better explain the motivating applications.
>
> **Minor comments:**
>
> *“Clarification on the use of \$K\$:”*
>
>
> In our study, \$K\$ denotes the number of classes, and \$[K]\$ represents the set of all class indices. Each class \$j\$ has a distinct upper bound \$\theta_j\$, and each arrival \$t\$ is associated with a label set \$\mathcal{J}_t\$, indicating its class membership. We have carefully reviewed the notation and confirmed that it is used consistently throughout the paper.
>
> *“Definition of \$\beta\$-NSW:”*
>
> NSW refers to Nash Social Welfare, a well-known concept in the literature, defined as the sum of the logarithms of the utilities across all classes. In the online setting, achieving the exact NSW is generally infeasible, so we focus on obtaining a \$\beta\$-approximation of it.
>
> *“FRAC-GFQ presentation in the main body:”*
>
> Thank you for the suggestion. We agree that including FRAC-GFQ in the main body would enhance clarity for readers. Unfortunately, due to space constraints, we were unable to include it in full. However, we will ensure it is properly referenced in the main text to minimize confusion.
>
> *“Description of algorithm 1:”*
>
> In the description of Algorithm 1, “Randomized” refers to the use of a randomized rounding scheme. “Set-Aside” indicates that the algorithm reserves a portion of the inventory for each class. Finally, “GFQ” highlights that the algorithm is designed to satisfy the Group Fairness by Quantity (GFQ) guarantee. We will clarify this further in the revised version.
>
> *“Pseudocode for the LiLA algorithm:”*
>
> We appreciate your suggestion. We will include the pseudocode for the LILA algorithm in the main body if space permits. Otherwise, we will place it in the appendix and add a clear reference in the main text to help readers understand the algorithm more easily.

---

> > ### Comment · Reviewer_fyob · 2025-08-04
> >
> > Thank you for your response. Including these additional clarifications in the final version of the paper sounds good, and I maintain my score above.

---

### Official Review · Reviewer_auby · 2025-07-03

**Clarity:** 2
**Significance:** 3
**Originality:** 3
**Rating:** 4
**Confidence:** 2

**Summary:**

The paper studies the online multiclass selection problem  OMcS) under two fairness constraints namely group fairness by quantity (GFQ) and $\beta$ proportionality. Unlike the usual setting, here the agents arrive online and make a request for 1 unit of finite indivisible resources for a price and the algorithm(seller) has to make an irrevocable decision to allocate the resource or reject the request. The goal is to maximize the utility while also adhering to the mentioned group fairness notion. Here the agents may belong to more than one protected group simultaneously and one allocation may lead to simultaneously satisfying multiple constraints.

The authors give
1) An optimal deterministic algorithm and a rounding based randomized algorithm for the problem with GFQ constraints. The rounding scheme is lossless and may be applied to other online settings
2) Rounding based randomized algorithm for problem the $\beta$- proportionality fairness constraint
3) A learning augmented framework to improve the algorithm in practice in presence of 'fair' advice.

The paper is theoretical in nature. However, authors have provided proof of concept experiments in the appendix.

**Questions:**

See Weakness and Suggestions

**Ethical Concerns:**

["NO or VERY MINOR ethics concerns only"]

**Final Justification:**

I have read the rebuttal. I believe the paper needs significant rewrite to be more readable. I will keep my score.

**Limitations:**

See Weakness and Suggestions

**Paper Formatting Concerns:**

See Weakness and Suggestions

**Quality:**

2

**Strengths And Weaknesses:**

Strengths:
1) The paper studies multi labeled group fairness in online setting with theoretical guarantees. The question is important both theoretically and practically and the paper can be simultaneously of interest to multiple communities.
2) The problem setting is clearly defined and there are multiple contributions in terms of algorithms and guarantees. I did not check the proofs in detail, but the paper appears sound in its arguments.
3) Proof of concept experiments are given in the appendix.

Weakness And Suggestions:
Though, as mentioned in the strengths, the problem setting is well defined and clear; my main concern with the paper is with the overall writing and structure of the paper. The paper is too dense to read and is not easily accessible.
I think the authors have tried to cram in a lot in the main body of the paper and have missed out on some more important things (in my opinion)
1. The most important thing I would like to see in the paper is more intuitive explanation of the algorithms, more insights and discussions on the definitions, algorithms and probably a section on proof ideas. For example, intuitively, what do the thresholds mean in theorem 3.1 and are constructed the way they are? what do the conditions signify in a practical setting? Some explanation of these kinds of things will make the paper much more readable.
2. Though the paper is theoretical in nature, since Neurips is a venue which caters to a broader audience, I think paper with strong theory with a good mix of empirical results will be better appreciated. I would suggest bringing some of the experiments to the main paper from the appendix.
3. I understand that there are space constraints for the main body of the paper, however I can suggest a couple of things
         i) Your related work is too detailed, make it crisper and move the bulk of it to the appendix
        ii)I think the main algorithms with the worst-case assumptions are themselves decent enough and the learning augmented framework may be mentioned in very brief in the main paper and rest of it can be moved to appendix.
4.  Do provide LiLA in the algorithm box format (may be in appendix). I find it difficult to follow it exactly in the current descriptive version

I do believe the paper is strong however in the current form I found it harder to parse. It is very mechanical and dry. I think these suggestions, if incorporated, might make it more readable and accessible to a broader audience.

---

> ### Author Rebuttal · Authors · 2025-07-31
>
> Thank you for your insightful review and constructive feedback. Below, we address your comments in a point-by-point manner:
>
> *"Intuitive Explanation and Insights:"*
>
> We fully agree that providing a more intuitive explanation of our algorithms, insights into the definitions, and discussion on threshold constructions can significantly enhance readability. The threshold functions employed in our algorithms are pivotal, as they determine whether incoming requests are accepted based on current utilization. Intuitively, the thresholds ensure balanced resource allocation by dynamically adjusting according to past decisions. In the revised manuscript, we will explicitly discuss the rationale behind threshold choices and the practical implications of their associated conditions, aiming to offer clear insights and facilitate better understanding.
>
> *"Inclusion of Empirical Results in Main Manuscript:"*
>
> We appreciate your recommendation to include empirical results within the main body of the paper. We initially placed these materials in the appendix due to page constraints. However, recognizing the importance of your suggestion, we will incorporate selected experimental evaluations into the main manuscript. Additionally, we have recently performed some more experimental results related to the caching problem and we plan to include part of these new empirical results in the revised paper..
>
> *"Adjustment of Related Work and Learning-Augmented Framework:"*
>
> Thank you for your valuable suggestion. We agree that adjusting the presentation of the paper can make it more rigorous and solid for a general audience. In the final version, we plan to include some experimental results directly in the main body and, if necessary, will move portions of the detailed related work section to the appendix. Additionally, we will provide a concise summary of the learning-augmented framework in the main manuscript, while the detailed discussion will be relocated to the appendix to maintain clarity and readability.
>
> *"Algorithm Box Format for LiLA:"*
>
> We agree that adding pseudocode for LiLA can enhance readability. We will include the pseudocode in an algorithm box format within the main body if additional pages become available. If this is not possible due to space constraints, we will ensure the pseudocode is clearly presented in the appendix. Thank you for the suggestion.

---

> > ### Comment · Reviewer_auby · 2025-08-05
> > **reply**
> >
> > Thanks for the rebuttal. I will keep my score.

---

### Official Review · Reviewer_k1BK · 2025-07-03

**Clarity:** 3
**Significance:** 3
**Originality:** 3
**Rating:** 5
**Confidence:** 2

**Summary:**

This paper tackles improving group fairness in the Online Multi-Class Selection (OMcS) problem, which allocates limited integer resources to sequentially arriving multi-class agents. To this end, the authors introduce a threshold-based algorithm and theoretically prove that it satisfies the group-fairness constraints while attaining the tight competitive ratio in utility maximization.

**Questions:**

Please refer to the weaknesses above.

1. Could you provide additional experimental results on other domains or datasets?

2. How about including some implementation details or a concise summary of the experimental setup and results within the main text?

**Ethical Concerns:**

["NO or VERY MINOR ethics concerns only"]

**Limitations:**

yes

**Quality:**

3

**Strengths And Weaknesses:**

-C1. The authors mathematically formalize the fundamental trade-off between group fairness and utility maximization in an online setting. On top of that, they propose a threshold-based algorithm that achieves a tight solution. As one of the pioneering works on this task, I believe that  this work will serve as an important foundation for future research.

-C2. The paper is well-written and logically structured, and the proposed algorithm is supported by rigorous theoretical guarantees, including tight competitive-ratio bounds.

-C3. Compared to the ADV baseline, the proposed method achieves more robust fairness guarantees while incurring only a marginal loss in total utility.

-W1. The current experiments are limited to a single task, limiting the reliability of empirical evaluations.

-W2 The current draft lacks implementation details, experimental setup, and results. They are included only in supplementary material.

---

> ### Author Rebuttal · Authors · 2025-07-31
>
> Thank you for your insightful review and valuable feedback. Below, we address your comments clearly and individually:
>
> *"Additional Experimental Results on Other Domains or Datasets:"*
>
> We appreciate your suggestion. We have recently conducted new experimental evaluations on an additional domain, specifically the network caching problem. In this scenario, a cache buffer with limited capacity is shared among multiple websites, and the goal is to strategically allocate this space fairly among diverse users. For this purpose, we utilized the Wikipedia Clickstream dataset, which contains hit statistics for webpages across multiple languages. Our objective in this experiment is to ensure fairness in cache allocation among users from different linguistic groups. Based on our new experimental results, we observed that fairness can be ensured at a moderate cost of efficiency, depending on how important the fairness guarantee is to the decision-maker. Additionally, we empirically evaluated the effect of the number of classes on the algorithm’s competitive ratio, which highlights that a higher number of classes can indeed make the problem more challenging. We will include these new experimental results in the final manuscript to broaden the scope and applicability of our findings.
>
> *"Implementation Details and Experimental Setup in the Main Text:"*
>
> We completely agree with your recommendation to incorporate a concise summary of our experimental setup and results within the main manuscript. Initially, these details were placed in the appendix due to page limitations. However, recognizing the importance of your feedback, we will integrate key implementation details and a brief overview of our experimental approach and results directly into the main body of the paper. Thank you for the suggestion.

---

### Official Review · Reviewer_ZEyX · 2025-07-07

**Clarity:** 3
**Significance:** 3
**Originality:** 3
**Rating:** 5
**Confidence:** 2

**Summary:**

Authors study the following problem setting.
- Agents arrive in a sequential patter onto the platform and query for an item (from the finite inventory). Each agent has an associated set of labels (categories that the agent belongs to), and a valuation for the item.
- The platform has to determine to either allocate or reject the agents' query as soon as they arrive. The platform is aware of the variance of valuations for different categories apriori, but does not know the distribution of labels of agents yet to arrive.

Authors attempt to solve a utility (summation of valuations where items get allocated to the agents) maximization under fairness constraints. They define competitive ratio of their proposed approach in form of a robust (min-max) formulation comparing the allocation output by their approach with that of optimal algorithm.

Authors define two fairness metrics (1) Group Fairness by Quantity (GFQ) (2) $\beta$-proportional Fairness (BPF). They propose an optimal deterministic algorithm for GFQ followed by a randomized variant. For BFP, authors propose another randomized st-aside algorithm and provide analysis for its performance.

Authors then comment on the overly pessimistic output of their proposed algorithm on account of the min-max formulation and propose a hybrid approach that probabilistically samples from either the output of their algorithm or from an "advice" allocation from a blackbox ML algorithm. Authors provide performance guarantees for the hybrid approach.

**Questions:**

1. Could the authors make Section 5 (learning augmented algorithm) clearer by creating sub-sections for limitation of Algorithm 3, athe proposed learning augmented algorithm (with pesudocode), the 2 definitions of fairness of allocation and analysis. [ I am leaning towards improving the score based on this relatively minor change]

2. Please include complexity analysis for each of the proposed approaches in the main manuscript.

3. [Minor] Authors have included experimental evaluation in the Appendix. I understand the limitation of page limit, but it would be valuable if the authors could include the evaluation in the main manuscript.

**Ethical Concerns:**

["NO or VERY MINOR ethics concerns only"]

**Final Justification:**

I am satisfied with the responses of the authors. Since there is no way to require authors to submitted a new manuscript in this stage, I choose to allow the authors to make these changes in the camera-ready version of the paper.

**Limitations:**

Yes

**Quality:**

3

**Strengths And Weaknesses:**

Strengths:

1. Overall, the presentation of content is very high quality. Even for a reader with very limited background in the related works, it was easy to parse and understand the problem setting and the contributions of the work.
2. I appreciated the inclusion of Section 5, wherein the authors acknowledge the limited practicality of their proposed approach and suggest an approach that overcomes some of those limitations.
3. Throughout the paper, authors place their mathematical results in context of the results from the related works and broader literature on the topic.

Weaknesses:

1. Section 5 could be made clearer by providing pseudocode.
2. Lack of complexity analysis.

---

> ### Author Rebuttal · Authors · 2025-07-31
>
> Thank you for your insightful review and constructive feedback. Below, we address your comments clearly and individually:
>
> *"Clarifying Section 5 (Learning-Augmented Algorithm):"*
>
> We agree that Section 5 can benefit from improved clarity. We will enhance this section by explicitly detailing how our proposed LiLA algorithm addresses the limitations of Algorithm 3, particularly regarding the handling of future uncertainties. Furthermore, we fully concur that including pseudocode for LiLA will substantially improve the readability and clarity of our approach. If additional pages become available upon acceptance, we will incorporate the pseudocode into the main text; otherwise, we will clearly present it in the appendix.
>
> *"Complexity Analysis:"*
>
> Thank you for highlighting this important aspect. Our proposed algorithm operates online and executes in \$O(1)\$ time per buyer arrival. Specifically, it calculates the fractional allocation  \$x_t\$ through a constant number of operations using predefined threshold functions for each class. Subsequently, the fractional allocation is rounded, also in \$O(1)\$ time, followed by making the final decision. Moreover, the memory complexity of our algorithm scales linearly with the number of classes \$K\$. The learning-augmented algorithm LiLA uses a convex combination of the predicted fair solution and the robust solution as the final decision, and thus it also takes \$O(1)\$ time per arrival. Therefore, our algorithm is both highly scalable and free from significant implementation bottlenecks. We will add discussions about the complexity of algorithms in the final manuscript.
>
> *"Inclusion of Experimental Evaluations:"*
>
> We appreciate your suggestion to integrate experimental evaluations from the appendix into the main manuscript. Initially, we placed these materials in the appendix due to page limitations. However, we fully agree that incorporating selected experimental evaluations into the main text would strengthen the manuscript’s structure and overall motivation. Additionally, we have recently conducted new experiments on the network caching problem, and we plan to include these new results in the main manuscript to further enhance its relevance and appeal to readers.

---

> > ### Comment · Reviewer_ZEyX · 2025-08-05
> >
> > Thank you for the response. Conditional on the changes recommended changes being made, I will increase my rating.

---

### Note · Authors · 2025-08-12

We sincerely thank the reviewers for their constructive feedback and valuable suggestions, which have helped us identify several ways to strengthen our work. In the final version of the paper, we will incorporate the following improvements:

1. We will provide detailed pseudocode for the proposed LiLA algorithm and expand its explanation to improve clarity and accessibility, as suggested by Reviewers **ZEyX**, **auby**, and **fyob**.

2. We will explicitly discuss the computational complexity and execution procedure of our algorithms in the main body to better highlight their practicality and scalability, as recommended by Reviewers **ZEyX** and **auby**.

3. We will move selected experimental results from the appendix to the main manuscript, as suggested by Reviewers **ZEyX** and **k1BK**.

We once again thank all the reviewers for their thoughtful and constructive input.

---

### Decision · Program_Chairs · 2025-09-17

**Decision:**

Accept (poster)

**Comment:**

The paper studies an online allocation problem with two different group fairness guarantees. In this problem, there are B available identical items. Agents arrive online and bid for one item. Each agent can belong to a number of subgroups. In one fairness definition, we want to allocate a certain minimum amount to each subgroup. In the second fairness definition (beta proportional fairness), we want to make sure that on average over all subgroups, no other allocation can achieve beta times higher utility. The paper proposes new algorithms based on first reserving the necessary items to satisfy the fairness requirement and then assign with time-dependent reserve price.

The reviewers generally appreciate the importance of the problem, especially allowing each agent to belong to many groups simultaneously. The randomized rounding technique is common in online assignment but has not been common in this literature. On the downside, the work is closely related to recent works where each agent belongs to only one group. The reviewers all put the paper above the bar.